# Spin-polarized oxygen evolution reaction under magnetic field

Xiao Ren[1,2], Tianze Wu [1,2,3], Yuanmiao Sun[2], Yan Li[1], Guoyu Xian[1], Xianhu Liu [4], Chengmin Shen[1], Jose Gracia [5], Hong-Jun Gao [1], Haitao Yang [1✉] & Zhichuan J. Xu [2,3,6✉]

The oxygen evolution reaction (OER) is the bottleneck that limits the energy efficiency of water-splitting. The process involves four electrons' transfer and the generation of triplet state $O_2$ from singlet state species ($OH^-$ or $H_2O$). Recently, explicit spin selection was described as a possible way to promote OER in alkaline conditions, but the specific spin-polarized kinetics remains unclear. Here, we report that by using ferromagnetic ordered catalysts as the spin polarizer for spin selection under a constant magnetic field, the OER can be enhanced. However, it does not applicable to non-ferromagnetic catalysts. We found that the spin polarization occurs at the first electron transfer step in OER, where coherent spin exchange happens between the ferromagnetic catalyst and the adsorbed oxygen species with fast kinetics, under the principle of spin angular momentum conservation. In the next three electron transfer steps, as the adsorbed O species adopt fixed spin direction, the OER electrons need to follow the Hund rule and Pauling exclusion principle, thus to carry out spin polarization spontaneously and finally lead to the generation of triplet state $O_2$. Here, we showcase spin-polarized kinetics of oxygen evolution reaction, which gives references in the understanding and design of spin-dependent catalysts.

[1] Beijing National Laboratory for Condensed Matter Physics and Institute of Physics, Chinese Academy of Science, Beijing, China. [2] School of Material Science and Engineering, Nanyang Technological University, Singapore, Singapore. [3] The Cambridge Centre for Advanced Research and Education in Singapore, 1 CREATE way, Singapore, Singapore. [4] Key Laboratory of Advanced Material Processing & Mold (Zhengzhou University), Ministry of Education, Zhengzhou, China. [5] MagnetoCat SL, Alicante, Spain. [6] Energy Research Institute @ Nanyang Technological University, Singapore, Singapore.
✉email: htyang@iphy.ac.cn; xuzc@ntu.edu.sg

The sluggish kinetics of oxygen evolution reaction (OER) is a major cause for the low efficiency in techniques, such as solar water splitting,[1] rechargeable metal-air batteries,[2] regenerative fuel cells,[3] and water electrolysis.[4,5] Exploring better catalysts for OER has become increasingly attractive in recent years. Non-precious 3d-transition metal oxides (TMOs), such as Fe-, Co-, and Ni-based oxides, are promising cost-effective catalysts.[6,7] Their catalytical activities are tunable as the diversity in oxides families affords numerous freedoms to tailor their physicochemical properties. Sabatier's principle, which qualitatively describes that the optimized catalytic activity when adsorbed species bind to the catalytic surface neither too strongly nor too weakly, led to the fundamental understanding of OER mechanisms and guided the subsequent design of highly active catalysts.[8,9] This principle was further supported by the findings that the OER activities of transition metal oxides correlate strongly with the $e_g$ occupancy, which is related to the binding strength between the metal and the oxygen species.[10–12] Some exceptions have been found not well fitted with the $e_g$ theory, which is partially resulted by the diverse and complicated magnetism in TMOs family.[13–16] Besides, the produced $O_2$ is in triplet ground state, where the frontier $\pi^*$ orbitals are occupied by two electrons with parallel alignment. In contrast, the ground spin state of reactant $OH^-/H_2O$ is singlet with all paired electrons.[17,18] The singlet states of the oxygen molecule were reported at an energy level of at least ~1 eV higher than its triplet state.[18,19] Thus, the magnetism of TMOs, related to the spin polarization, should be influential on the kinetics of OER.[20,21] It is reasonable to consider that the active sites with suitable thermodynamic paths for OER should allow a way to facilitate the spin alignment in the product. As suggested by recent theoretical studies by J. Gracia,[22–24] the spin-polarized electrons in catalysts promote the generation of parallel spin aligned oxygen by quantum spin-exchange interactions (QSEI), which further promote the OER kinetics. Therefore, facilitating the spin polarization should be an effective strategy for improving OER efficiency. Ron Naaman and co-works reported that the application of the chiral-induced spin selectivity effect to product the polarized electron. This spin polarization transferred is the origin of a more efficient oxidation process in which oxygen is formed in its triplet ground state.[25–27] It has been pointed out by J. Gracia et al. that theoretically photosystem II can act as a spin-controlling gate to govern the charge and spin transport during the OER process,[28] which offers a favoured thermodynamic path for $O_2$ evolution. Besides the extrinsic spin polarizer, the ordered magnetic moment structure in ferromagnetic materials can create intrinsic spin filtering for highly spin-polarized electrons. The spin filtering effect originates from the exchange splitting of the energy levels in the conduction band of a magnetic insulator.[29] Most recently, José Ramón Galán-Mascarós et al. reported an external magnetic field, applied by a permanent magnet, enhances the OER activity of magnetic oxides in alkaline.[30] It opens a new strategy to manipulate the spin polarization in magnetic oxide catalysts for promoting the OER and encourages more detailed studies to understand how the magnetic field induced spin polarization affects the OER process.

In this work, we report an investigation on the key kinetics change on the ferromagnetic $CoFe_2O_4$ catalyst under the magnetic field. The ferromagnetic $CoFe_2O_4$ catalyst works as a spin polarizer under the magnetic field. We have found that the spin-polarized kinetics of OER starts at the first electron transfer step, where ferromagnetic exchange happens between the ferromagnetic catalysts and the adsorbed oxygen species (reactants) under the principle of spin angular momentum conservation. Without the magnetic field, the Tafel slope of $CoFe_2O_4$ is identical and equal to circa 120 mV/decade, which indicates the

first electron transfer step is rate-determining step (RDS) and no electron transfer occurring before the RDS. Under the magnetic field, the Tafel slope decreases to circa 90 mV/decade, indicating the number of electron transfer is ~0.5 and a mixed RDS involving the first electron transfer step and second steps. Such a phenomenon cannot be observed in the catalysts without ferromagnetic orderings under the same condition. The results indicate that the key step of spin-polarized OER is the first electron transfer step in OER, where the spin-polarized process via exchange hopping can be facilitated under the magnetic field. As a consequence, the first electron transfer is no longer the RDS. After a facilitated spin-polarized ferromagnetic exchange of electrons, the adsorbed O species will overall settle on the fixed spin direction. Due to the Hund Rule and Pauli Exclusion Principle, the follow-up electrons' transfer needs to carry out spin polarization spontaneously and finally lead to the generation of triplet state oxygen. Overall, we showcase the key kinetics information in OER under a magnetic field, which influences the micro- and macroscopic spin polarization and spin transport. This finding will be helpful for further development of magnetic field assisted OER-enhancing strategy and related catalysts.

## Results

**Magnetic and electrochemical characterizations**. We begin with the discussion of the magnetic properties of the employed catalysts, $CoFe_2O_4$, $Co_3O_4$, and $IrO_2$. The study will determine the suitable magnetic field and weather a global aligned magnetic moment can be achieved. The study of magnetic property reveals the optimal strength of the applied magnetic field for the alignment of the magnetic moment in ferromagnetic $CoFe_2O_4$. The $CoFe_2O_4$ and $Co_3O_4$ were prepared by a modified solid-state chemistry method as previously reported.[11] X-ray powder diffraction characterization was performed to confirm their crystal structures. The diffraction patterns match well with the standard patterns without impurity peak found (Supplementary Fig. 1 and Supplementary Table 1). As depicted in Fig. 1a, $CoFe_2O_4$ gives a hysteresis loop in an enlarged manner, indicating a room-temperature ferromagnetic behavior with a saturation magnetization (Ms) of 44 emu·g$^{-1}$. The $Co_3O_4$ and $IrO_2$ samples with tiny susceptibility ($\chi$) of $3.07 \times 10^{-5}$ and $0.51 \times 10^{-6}$, respectively, at 300 K show antiferromagnetic or paramagnetic behaviors, respectively. The detailed magnetic data are summarized in Supplementary Table 1. The cyclic voltammetry (CV) of those catalysts were then measured with and without a constant magnetic field of 10,000 Oe under alkaline condition (see Methods for details). As shown in Fig. 1b, c, d, the OER performance of the ferromagnetic $CoFe_2O_4$ is promoted obviously under the magnetic field while the changes in non-ferromagnetic catalysts $Co_3O_4$ and $IrO_2$ are negligible. When a strong enough magnetic field (higher than the coercivity) is applied to a ferromagnetic material, the magnetic moment will (macroscopically) align along with the direction of the external magnetic field. The ferromagnetic (long-) ordered material as spin polarizer is an extended selective spin-filter for electron transfer during catalysis. The generation process of polarized electrons has been illustrated in Fig. 1e.

It should be noticed that the use of magnetic fields in water electrolysis has been studied in the past,[31–36] in which the mass transport in the electrochemical process was found to be affected by the Lorentzian movement, i.e. the diffusion of regents and the release of the generated gas bubbles are promoted. However, in this study, some evidence has excluded such effect from mass transport as a main contributor to OER enhancement under the magnetic field. First, the improvement was not observed on

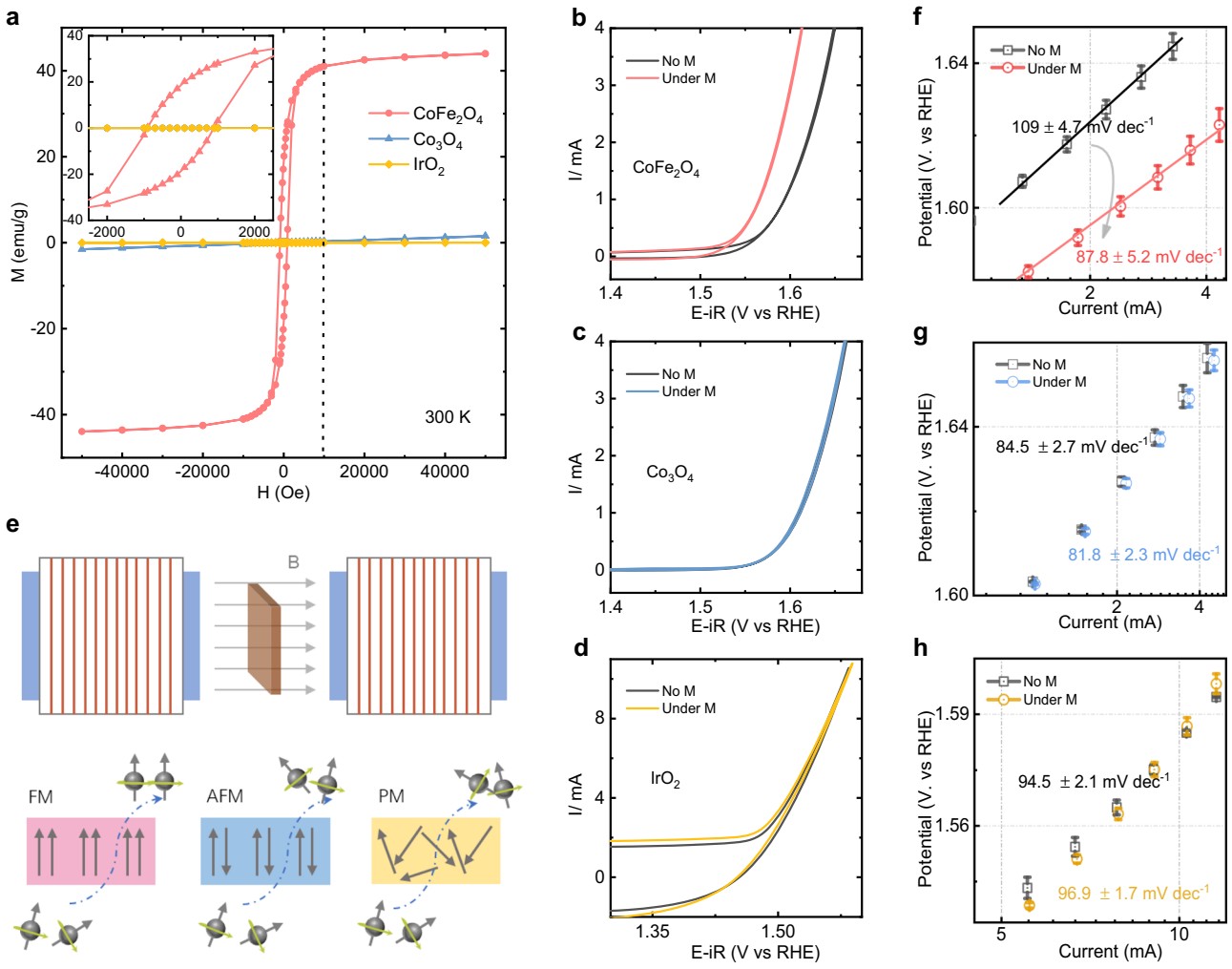

**Fig. 1 Spin polarization promotes OER. a** Magnetic hysteresis loops of CoFe$_2$O$_4$, Co$_3$O$_4$, and IrO$_2$ powders at room temperature (300 K) and the magnified graph inset in the top left of this panel. (Here, IrO$_2$ powder is a commercial catalyst (bulk, Premetek). Cyclic voltammetry (CV) of CoFe$_2$O$_4$ (**b**), Co$_3$O$_4$ (**c**), and IrO$_2$ (**d**) catalysts at a scan rate of 10 mV/s in O$_2$-saturated 1 M KOH with and without a constant magnetic field (10,000 Oe). **e** The schematic of the generation of the polarized electron under a constant magnetic field. The Tafel plots of CoFe$_2$O$_4$, (**f**) Co$_3$O$_4$ (**g**), and IrO$_2$ (**h**) catalysts with and without a constant magnetic field (10,000 Oe). The error bar represents three independent tests.

non-ferromagnetic catalysts Co$_3$O$_4$ and IrO$_2$ with the effect of Lorentzian movement on mass transport. Second, we also tested the OER performance of Co(acac)$_2$ and Fe(acac)$_3$ with and without a constant magnetic field (as shown in Supplementary Fig. 2). Nearly no difference can be observed. It also should be noted that OH$^-$ and H$_3$O$^+$ in aqueous solution do not move physically, but by sequential proton transfer, known as Grotthuss mechanisms[37] (Supplementary Fig. 3). That means the influence of Lorentz force on the physical movement of ions OH$^-$ or H$_3$O$^+$ is negligible. Thus, the effect from the mass transport under the external magnetic field should have little contribution to the observed OER enhancement of the ferromagnetic CoFe$_2$O$_4$. Besides, the electrical resistance of magnetic materials can be affected by the magnetization, which is through the scattering of electrons on the magnetic lattice of the crystal.[38–40] However, the difference in the conductivity at room temperature under 10,000 Oe is about 3% for insulator CoFe$_2$O$_4$ with 3.86 × 10$^{-5}$ S/m,[40–42] which does not cause a significant difference in the electrode's conductivity. This is because acetylene black carbon (AB) with 500 S/m as a conductive mediator is mixed with those oxide catalysts for their application as the electrode,[43] which dominants the electron conduction.

**No surface restructuring in OER.** It is generally recognized that some Co-based perovskites and spinels undergo operando surface reconstruction to form active Co (oxy) hydroxides in alkaline conditions to promote OER.[44–46] In our case, there are no changes in OER performance of CoFe$_2$O$_4$ during CA tests in 1 M KOH for 1 h shown in Supplementary Fig. 4, indicating CoFe$_2$O$_4$ is stable without noticed surface reconstruction during the OER process. The high-resolution transmission electron microscope (HRTEM) was further used to rule out the possible interference from surface reconstruction of catalysts during the OER. It has been found that the spinel crystal structure of CoFe$_2$O$_4$ remained after the electrochemical treatment (Supplementary Fig. 5), which is consistent with what has been reported previously.[47] The aberration-corrected STEM provides direct atomic imaging and confirms that the well-crystalline feature reserved from the surface to bulk (Fig. 2a, b). The HADDF line profile shows the same bond length of Co-O in bulk and surface, which verifies no surface reconstruction (Fig. 2c). Raman technique was then performed to study the inhomogeneity evolution in the near-surface region. The Raman spectra of cubic structures (Fd-3m) CoFe$_2$O$_4$ before and after OER are presented in Fig. 2d. In the top curve, peak maxima at 603 and 666 cm$^{-1}$ are due to

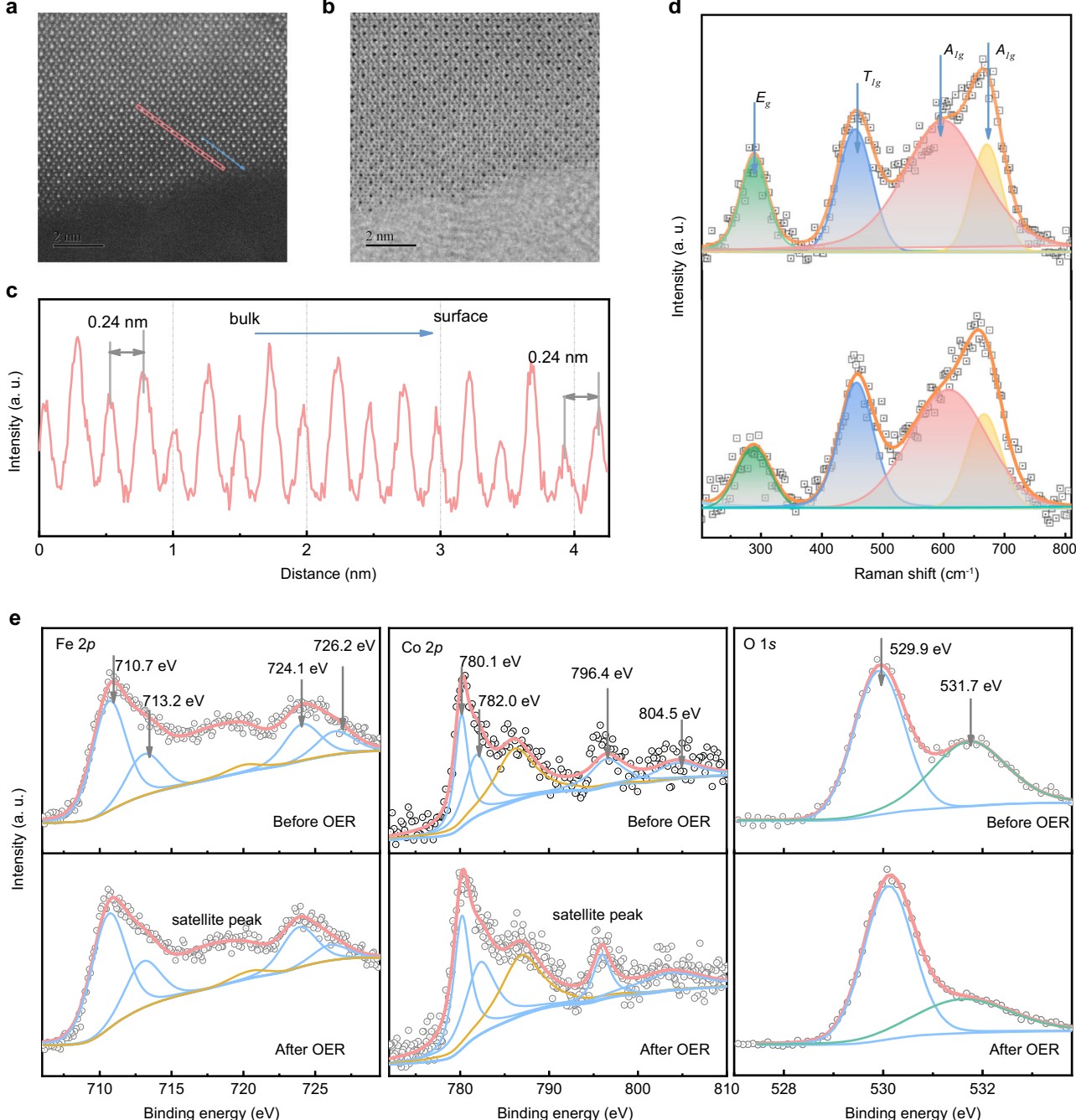

**Fig. 2 No surface restructuration on CoFe$_2$O$_4$.** HADDF (**a**) and ABF (**b**) images of CoFe$_2$O$_4$ after OER. The line profiles of HADDF (**c**) acquired at the pink line rectangular zone. **d** Raman spectra of CoFe$_2$O$_4$ before and after OER. **e**The Fe 2$p$, Co 2$p$, and O 1$s$ XPS spectra results of CoFe$_2$O$_4$ before and after OER.

$A_{1g}$ symmetry involving symmetric stretching of oxygen atom with respect to the metal ions in tetrahedral sites. The other low-frequency phonon modes are due to metal ions involved in octahedral sites, i.e. $E_g$ and $T_{1g}$. The assignment of these phonon modes was carried out in accordance with the literature.[48] After electrochemical treatment (bottom curve), no changes in the vibrational modes were observed, which proves once again that there is no surface reconstruction in OER. X-ray photoelectron spectroscopy (XPS) was also performed to study the surface chemical states of CoFe$_2$O$_4$ before and after the OER. As shown in Supplementary Fig. 6, the XPS survey spectra confirm the coexistence of Co, Fe, and O in the samples. Figure 2e shows

the 2$p$ orbital of Fe, Co, and 1$s$ orbital of O. The XPS of Fe 2$p$ core level presents two pairs of peaks: Fe$^{3+}$ 2$p_{3/2}$ at about 710.7 eV and 713.2 eV; Fe$^{3+}$ 2$p_{1/2}$ at about 724.1 eV and 726.2 eV. The doublets in samples can be ascribed to Fe$^{3+}$ in octahedral sites and Fe$^{3+}$ in tetrahedral sites, respectively. The two peaks of Co 2$p$ with the binding energy of 780.1 and 782.0 eV are ascribed to Co$^{2+}$ ions in octahedral sites and Co$^{2+}$ ions in tetrahedral sites. The main peaks of O 1$s$ at 529.6 eV are recognized as oxygen ions, which are all associated with a "−2" formal charge.[49] Compared to the spectra before and after the OER, these peaks remain unchanged in location, indicating no surface reconstruction.

**Spin-polarized kinetics of OER.** Oxygen evolution reaction is authenticated a four-step reaction with each step accompanied by an electron transfer. The Tafel plots are widely regarded as a generalized kinetics theory for electron transfer reactions. The Tafel equation presents the relationship between the Tafel slope and the exchange current density:

$$\eta = -\frac{2.303\,RT}{\alpha F} \times \log i_0 + \frac{2.303\,RT}{(\alpha + n)F} \times \log i \qquad (1)$$

where the Tafel slope equals to $2.303RT/[(\alpha + n)F]$ ($i_0$ is the exchange current density, R is the universal gas constant, T is the absolute temperature, F is the Faraday constant, n is the number of electrons transferred before RDS, and $\alpha$ is the charge transfer coefficient and usually assumed to be 0.5).[50,51] Ideally, the Tafel slope tells the information of reaction kinetics. For example, the Tafel slope is 120 mV·dec$^{-1}$, which indicates the first electron transfer step is the RDS because there is no electron transfer before the RDS. If the second step is the RDS, the Tafel slope will decrease to 40 mV·dec$^{-1}$ with an electron transfer number of 1. The change of the Tafel slope is often reputed as an indication of the change of reaction mechanism. As shown in Fig. 1f, the Tafel slope of CoFe$_2$O$_4$ is about 109 ± 4.7 mV·dec$^{-1}$ and that indicates the first electron transfer from the adsorbed OH$^-$ is the RDS without the magnetic field. But, after applying a constant magnetic field, the Tafel slope decreases to circa 87.8 ± 5.2 mV·dec$^{-1}$, indicating the number of electron transfer is about 0.5 and a mixed RDS involving the first electron transfer step and second steps. Furthermore, we have carried out OER measurements of CoFe$_2$O$_4$, Co$_3$O$_4$, and IrO$_2$ under different temperatures as shown in Fig. 3a. We first noted that the OER performance of catalysts is getting better as the reaction temperature increases. This is probably because that the rate constant of the reaction will increase as the reaction temperature increases, which can promote this reaction based on the transition state theory[52]. More importantly, the OER performance of the ferromagnetic CoFe$_2$O$_4$ is promoted under the magnetic field at various temperature. However, the positive influence of the magnetic field on the OER performance of CoFe$_2$O$_4$ is decreased as the reaction temperature increases. The corresponding Tafel slopes are shown in Fig. 3b. At room temperature, the Tafel slope of CoFe$_2$O$_4$ is about 106 mV·dec$^{-1}$ without the magnetic field. After applying a constant field, the Tafel slope decreases to circa 82.8 mV·dec$^{-1}$. As the temperature increases, the positive influence of the magnetic field became not that remarkable. This is because the arrangement of magnetic moments of catalyst will be thermally disturbed. The ferromagnetic ordering in the catalyst gets disturbed and thus a certain degree of demagnetization at high temperature occurs, which lead to the decreased influence of the magnetic field on OER. We also note that the Tafel slope of CoFe$_2$O$_4$ have a slight favorable change as temperature increases, which maybe because the interaction between two M-O unites mechanism occurs at high temperature.[53,54] Thus, the key step in spin-polarized OER is the first electron transfer step in FM CoFe$_2$O$_4$, where the adsorbed OH$^-$ is difficult to deprotonate and transfer the electron. However, the change of Tafel slopes was not observed in the non-ferromagnetic catalysts under the same condition.

The electron transfer at the catalytic interface depends on the transition probability, which is associated with the wavefunction integral between OH$^-$ and the active site. As revealed by our previous work, the octahedral sites are mainly responsible to the OER[55]. The extended X-ray absorption fine structure (EXAFS) showed the perfect inverse spinel structure of CoFe$_2$O$_4$ (Supplementary Fig. 7). The Fe$^{3+}$ ions distribute equally in octahedral and tetrahedral sites and Co$^{2+}$ ions distribute in octahedral sites. We further calculated the effective magnetic moment ($\mu_{eff}$) of

CoFe$_2$O$_4$ to be about 3.44 $\mu_B$ by Curie−Weiss fitting (Supplementary Fig. 8). The $\mu_{eff}$ for CoFe$_2$O$_4$ is very close to the idea value of the inverse spinel.[56] Thus, the Co$^{2+}$ ions in octahedral sites contribute to the effective ferromagnetic moment. Those results are consistent in previous experimental work.[57] Considering that only Co in octahedral sites contribute the effective magnetic moment, the magnetic field enhanced OER should mainly happen on the Co sites. Thus, we studied the Co sites as the active sites in this work. For a ferromagnetic (FM) catalyst, the orbitals of the FM oxides create an intrinsically degenerate spin-polarized metallic state that optimizes the wavefunction based on the inter-atomic reduction of the electron–electron repulsion. DFT calculations were performed to explore the different elctron structure of CoFe$_2$O$_4$ under an applied magnetic field (The computational details are shown in the Supplementary Information). As shown in the projected density of states (PDOS) of CoFe$_2$O$_4$ (Fig. 4a), there is more overlap between the line of M-3d and the line of O-2p after spin alignment, which indicates the 3d-2p hybridization of the CoFe$_2$O$_4$ become stronger[58] after spins are aligned. As well, compared to the CoFe$_2$O$_4$ with anti-parallel couplings, the CoFe$_2$O$_4$ with spin alignment has a higher spin density on the oxygen atoms (Fig. 4b). The calculation indicates that the magnetic moment of the ligand hole in CoFe$_2$O$_4$ is 0.059 $\mu_B$ without spin alignment and is 0.188 $\mu_B$ with spin alignment, which indicates a FM ligand hole in CoFe$_2$O$_4$. A concomitant increment of the 3d-2p hybridization associate with FM ligand holes will facilitate spin-selected charge transport and optimize the kinetics of the spin-charge transfer in the three-phase interface.[43,59] Thus, the dominant FM exchange between the ferromagnetic catalyst and the adsorbed oxygen species (reactants) will happen (Fig. 4c and Supplementary Figure 9) with smaller electron–electron repulsion, which induce spin-dependent conductivity and decrease the rate-limiting bonding energies, making that the first electron transfer is no longer the RDS. We further prepared the Pourbaix diagram of CoFe$_2$O$_4$ as shown in Fig. 4d, which show that the surface termination of CoFe$_2$O$_4$ is oxygen termination under OER conditions. The reaction started between a ligand oxygen on the surface and the adsorbed oxygen species (OH$^-$), and the "first" electron transfer step is O* + OH$^-$ → *OOH + e$^-$. The spin-related OER mechanisms show in Fig. 4e. The FM CoFe$_2$O$_4$ with FM ligand hole will form oxygen termination with fixed spin direction. The first electron transfer process led to the generation of O(↓)$^-$, that is, the first electron transfer step is spin-polarization process to form the triplet state intermediate O(↓)O(↓)H species with a lower barrier (Supplementary Fig. 10). Consequently, the triplet state intermediate O(↓)O(↓)H species will prefer to generate the triplet state O$_2$. We also conducted a DFT study on the free energies of OER steps on the (111) surface of CoFe$_2$O$_4$ with and without spin alignment. Please be noted that here the topmost layer of the slab model is fully relaxed in the calculations since there is little difference between the one-layer-relaxed model and the tww-layer-relaxed model, which can be also found in literature[60] The calculation model of CoFe$_2$O$_4$ is shown in Supplementary Fig. 14. The (111) surface is chosen because the TEM investigation found the surface is rich in (111) and there is no remarkbale change on the surface after OER (Supplementary Fig. 11). The energy diagram for these two paths at 1.23 V (vs RHE)[61,62] to produce triplet oxygen is shown in Fig. 4f. The active sites with spin alignment are more thermodynamically favourable to OER, if they associate with ferromagnetic ligand holes,[59] and the overpotential of producing triplet oxygen is reduced by 390 mV compared to that without aligning spin. The coordinated inter-atomic aligned spin on active sites plays an important role in optimizing the spin-dependent reaction coordinates.

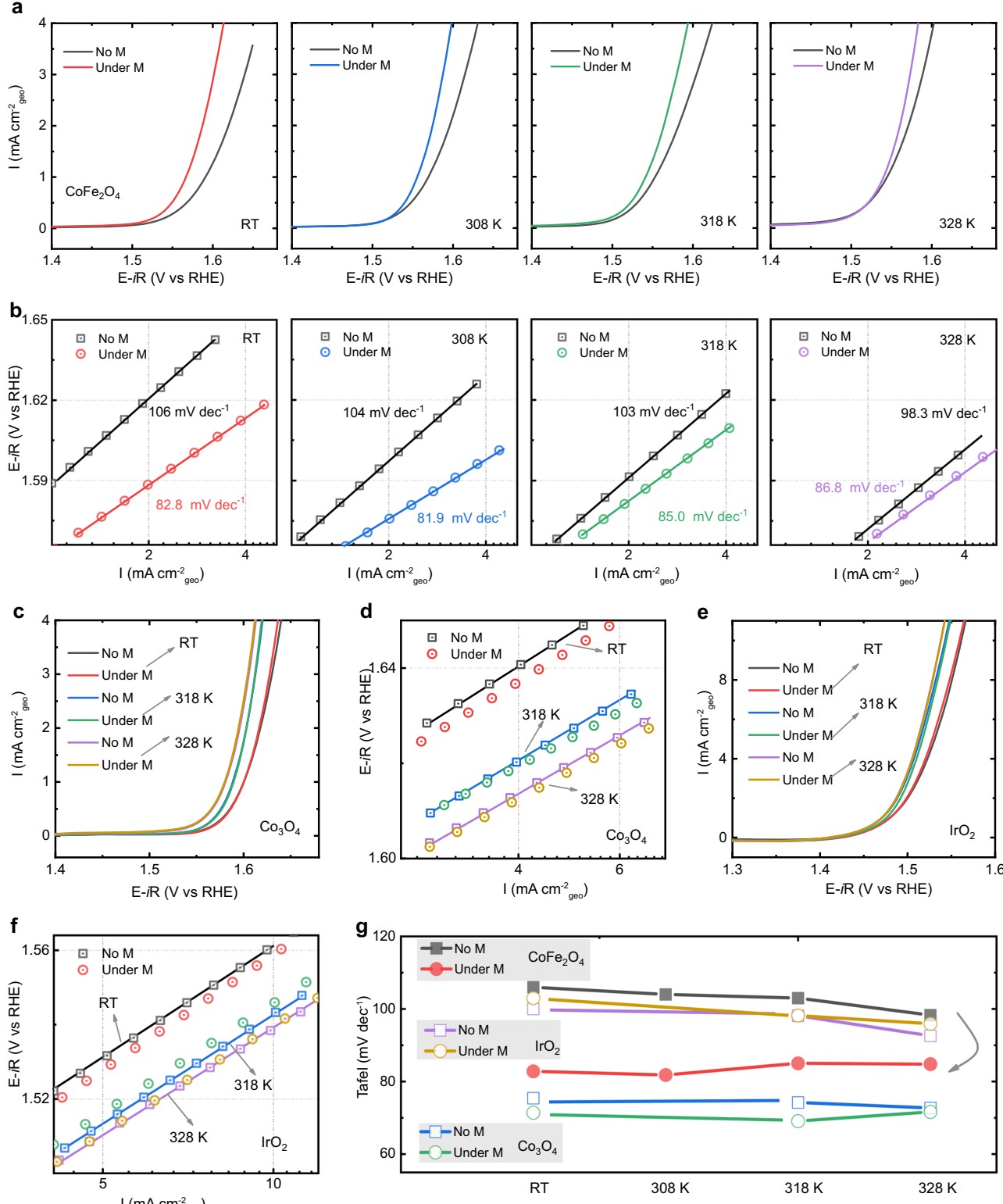

**Fig. 3 OER under the different temperature. a** LSV curves of CoFe₂O₄ at a scan rate of 10 mV/s in O₂-saturated 1 M KOH with and without a constant magnetic field (10,000 Oe) under the different temperatures (room temperature (RT): ~303 K, 308 K, 318 K, and 328 K). The corresponding Tafel plots are shown in **b. c** LSV curves of Co₃O₄ with and without a constant magnetic field (10,000 Oe) under the different temperatures (room temperature (RT): ~303 K, 313 K, and 323 K). The corresponding Tafel plots are shown in **d. e** LSV curves of IrO₂ with and without a constant magnetic field (10,000 Oe) under the different temperatures (room temperature (RT): ~303 K, 313 K, and 323 K). The corresponding Tafel plots are shown in **f**. Tafel slopes at various temperatures are summarized in **g**.

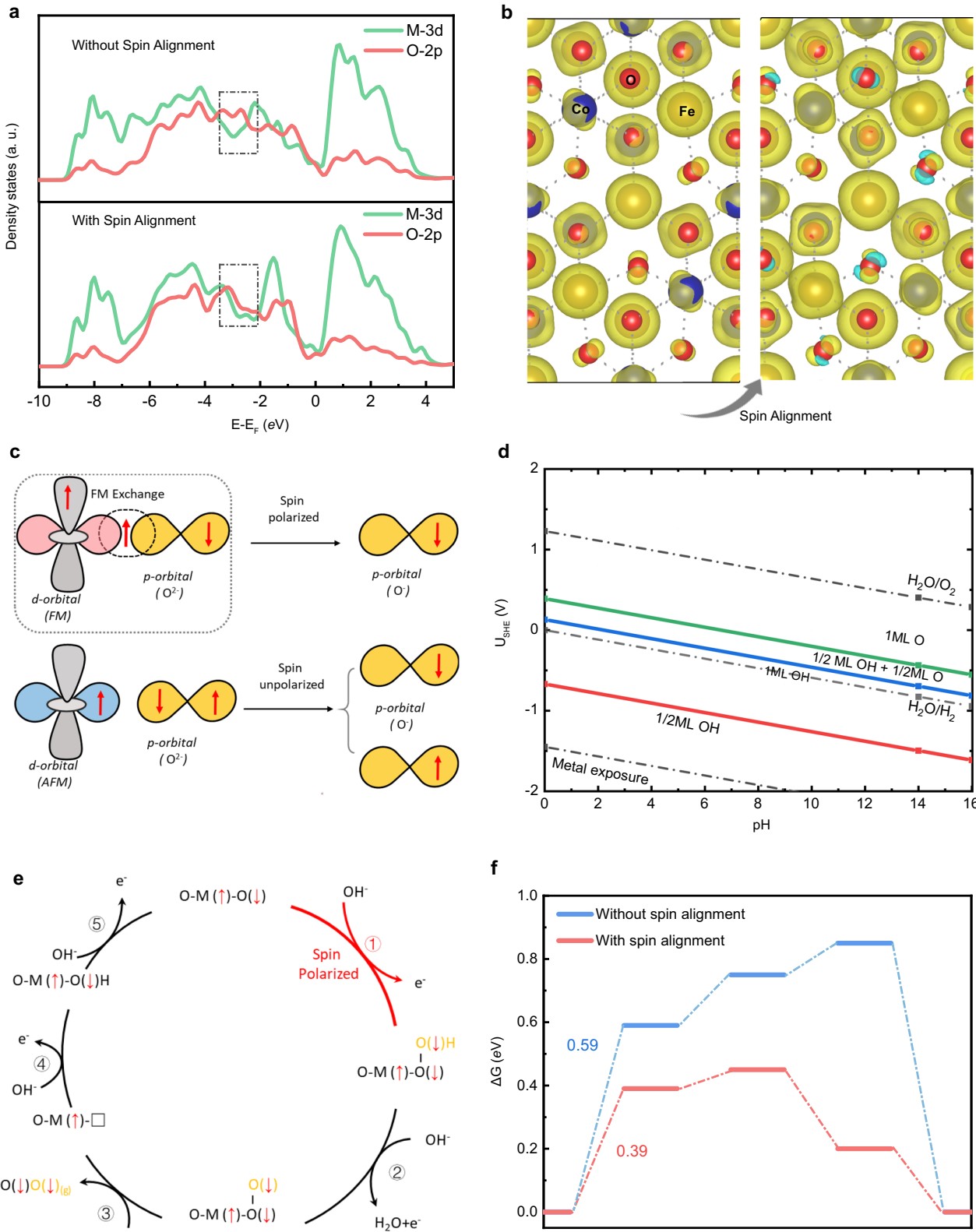

**Fig. 4 Spin-polarized OER. a** The projected density of states (PDOS) of $CoFe_2O_4$ without and with spin alignment. **b** The spin density for $CoFe_2O_4$ with and without spin alignment. **c** Schematic of spin-exchange mechanism for OER. The first electron transfer step is promoted by spin polarization through the FM exchange (QSEI), which gives smaller electronic repulsions and makes the adsorbed O species have a fixed spin direction. **d** The calculated Pourbaix diagram of the (111) surface of $CoFe_2O_4$. **e** The spin-polarization mechanisms in OER with starting from the step of $O* + OH^- \rightarrow *OOH + e^-$ step. **f** The free energy diagram of OER at 1.23 V (vs. RHE)[61,62] with and without the spin alignment on the (111) surface of $CoFe_2O_4$ toward triplet oxygen production.

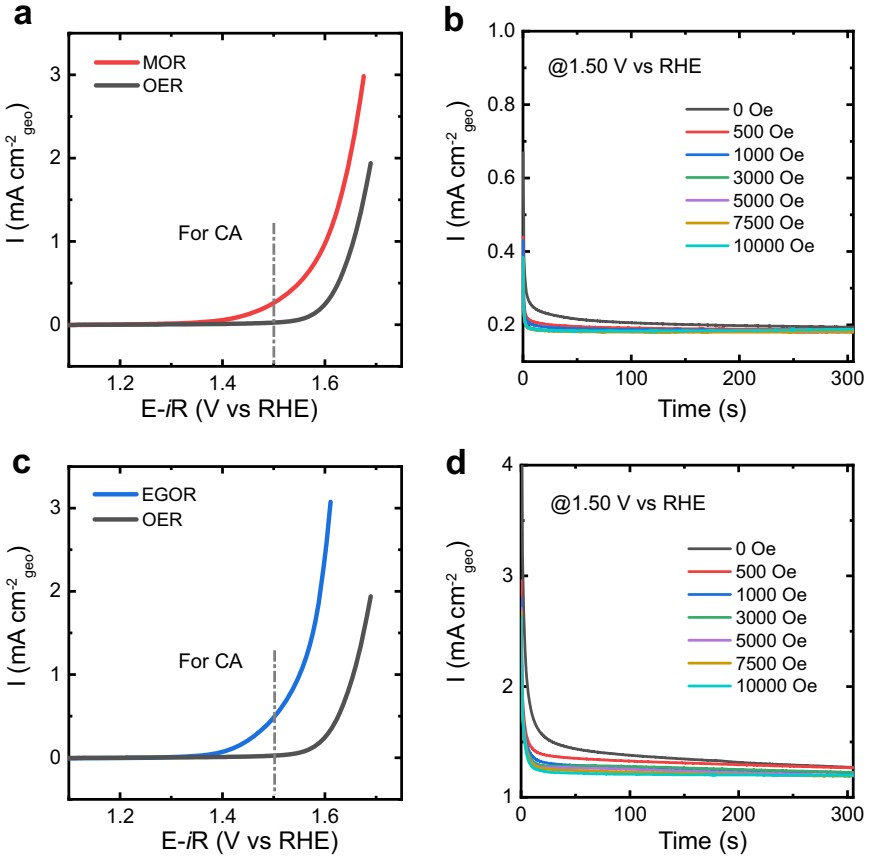

**Fig. 5 No influence on MOR and EGOR. a** Linear sweep voltammetry (LSV) curves of MOR at a scan rate of 10 mV/s in 1 M KOH in the presence of 1 ml methanol. **b** Chronoamperometry (CA) experiments of MOR were performed at a constant potential of 1.50 V (versus RHE). **c** LSV curves of EGOR at a scan rate of 10 mV/s in 1 M KOH in the presence of 1 ml ethylene glycol. **d** The CA curves of EGOR at 1.50 V vs. RHE.

It is worth noting that OER requires the generation of paramagnetic $O_2$ molecules starting from diamagnetic species ($OH^-$ and $H_2O$). For a reaction involving non-magnetic molecules only, not impact significantly on the reaction kinetics. We investigated the methanol oxidation reaction (MOR) and ethylene glycol oxidation reaction (EGOR) on $CoFe_2O_4$ under the magnetic field. Here, changes in FM catalyst conductivity caused by magnetic fields can be ignored because that AB mediator is also mixed with $CoFe_2O_4$ for their application as the electrode, which dominant the electron conduction. Figure 5 reveals that there is no remarkable difference in these reactions under the magnetic field. This is because the reactants, intermediates, and the products in these reactions are diamagnetic and there is no spin-selected electron transfer between the active metal site and the adsorbed reaction species.

**The effect of gradient magnetic field, remanence, and demagnetization**. It is known that for ferromagnetic materials, the magnetic moment pertains to the spin. The more ordered its magnetic moment is, the higher the degree of positive spin polarization is. As observed in the initial magnetization curve of $CoFe_2O_4$ (Fig. 6a), its magnetic moments become more orderly as the magnetic field increases, before reaching the saturation field. We then investigated the effect of the gradient magnetic field on OER activity. We carried out a series of CA measurements under the different magnetic field strength at a constant potential of 1.66, 1.66, and 1.56 V (versus RHE) for $CoFe_2O_4$, $Co_3O_4$, and $IrO_2$, respectively (Fig. 6b). It can be seen that the current density of the ferromagnetic catalyst $CoFe_2O_4$ increases with the increase of the magnetic field strength. For non-ferromagnetic $Co_3O_4$ and

$IrO_2$, there is almost no change when changing the field strength. The increment of the current density is summarized in Fig. 6c. The increase of spatial spin polarization related to the degree of magnetization shows a positive correlation with the enhancement of the OER for ferromagnetic $CoFe_2O_4$. We also measured the LSV curves of all oxides before and after the CA experiment. As seen in Supplementary Fig. 12, the OER performance ferromagnetic $CoFe_2O_4$ can be further improved after the CA test under magnetic field, but not for non-ferromagnetic $Co_3O_4$ and $IrO_2$. An interesting finding is that the OER performance of $CoFe_2O_4$ remains even after the magnetic field is removed (Fig. 6d). This is because the magnetic moment is still aligned in magnetized $CoFe_2O_4$ (Fig. 6e) after removing the magnetic field, which persists as the spin polarizer to create spin polarization. This is an important fact to make clear that the enhancement is due to the indirect (strong) QSEI, and not due to weak direct spin-spin interactions from the external field, a typically conceptual error. More interestingly, when the magnetized $CoFe_2O_4$ was demagnetized using an oscillating magnetic field (Fig. 6g), the OER performance of $CoFe_2O_4$ reverted to the initial value before the field was applied. The Tafel slope of $CoFe_2O_4$ is back to 120 mV dec$^{-1}$, indicating the first electron transfer of the adsorbed $OH^-$ is again the RDS, same as the status without the magnetic field. Based on the above results, we can confirm that the spin polarization facilitated OER is reversible and adjustable.

## Discussion

It is found that ferromagnetic $CoFe_2O_4$ serves as the spin polarizer facilitates the spin polarization under a constant magnetic field. The increase of spatial spin polarization shows a positive

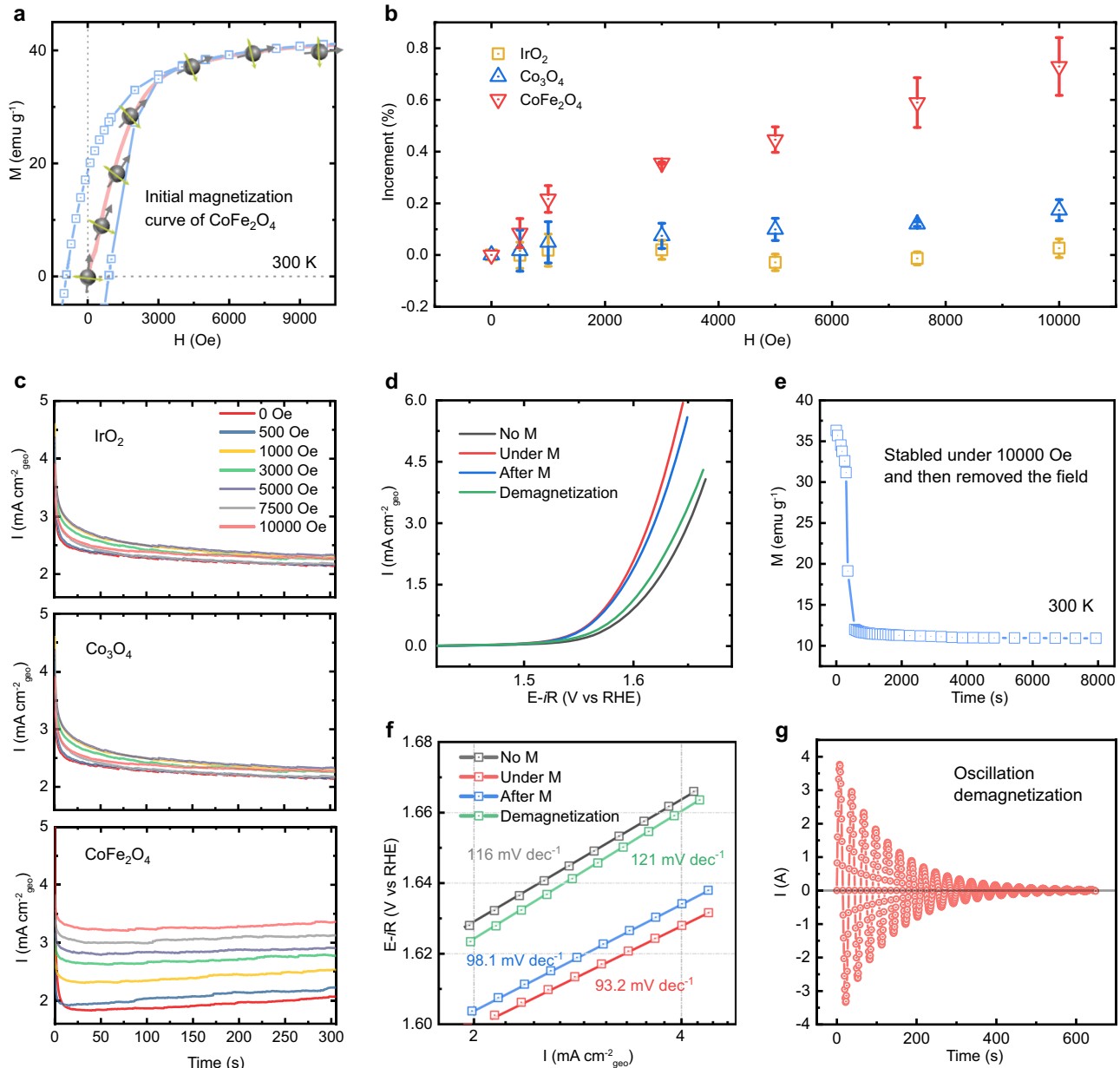

**Fig. 6 The effect of gradient magnetic field, remanence, and demagnetization. a** Initial magnetization curve of CoFe$_2$O$_4$. **b** CA test in 1 M KOH under the different magnetic field strength (0, 500, 1000, 3000, 5000, 7500, and 10000 Oe) at a constant potential of 1.66 V versus RHE for CoFe$_2$O$_4$, Co$_3$O$_4$, and 1.56 V versus RHE for IrO$_2$. **c** The increment of the current density under different magnetic field strength. It was calculated by the following equation: Increment (%) = $(j_M - j_{M=0})/j_{M=0}$; $j_M$ is the chronopotentiometry current density values obtained under the applied magnetic fields (0, 500, 1000, 3000, 5000, 7500, and 10000 Oe). The error bar represents three independent tests. **d** LSV curves of CoFe$_2$O$_4$ at a scan rate of 10 mV/s in O$_2$-saturated 1 M KOH with and without a constant magnetic field (10,000 Oe), after the magnetic field removed (after M), and after demagnetization. The corresponding Tafel plots are shown in **f**. **e** The magnetization of CoFe$_2$O$_4$ after removing a constant magnetic field of 10,000 Oe. **g** The curve of demagnetization for CoFe$_2$O$_4$.

correlation with the enhancement of spin transport (selection) during OER. We have found that the Tafel slope of overall ferromagnetic CoFe$_2$O$_4$ switched from ~120 to ~90 mV·dec$^{-1}$ after applying a magnetic field. It indicates the change of the RDS of OER reaction under an external magnetic field, i.e. the first electron transfer step is no longer the RDS. The spin-polarized electron exchange between the ferromagnetic CoFe$_2$O$_4$ and the adsorbed oxygen species (reactants) for the first electron transfer is ferromagnetic-exchange-like under the principle of spin angular momentum conservation, which leads to faster reaction kinetics

for the first electron transfer step. In contrast, such a phenomenon was not observed on non-ferromagnetic catalysts. The findings imply that the conservation of the total spin on the active sites during OER is an important concept, which applies quantum spin-exchange interactions to optimize reaction kinetics. The kinetic improvement maintains after the removal of the external magnetic field. The demagnetization can bring the activity back to that before magnetization. This work provides new understandings of the effect of an external magnetic field on the OER activity of a ferromagnetic catalyst.

## Methods

**Material synthesis**. Spinel $CoFe_2O_4$ oxides were synthesized by a modified conventional solid-state chemistry method as described elsewhere[45] with $Fe(NO_3)_2$ and $Co(NO_3)_2$ as precursors. 9 mmol mixture of $Fe(NO_3)_3 \cdot 9H_2O$ (Alfa Aesar) and $Co(NO_3)_2 \cdot 6H_2O$ (Sigma–Aldrich) was dissolved in 15 mL of DI water, followed by stirring and vaporizing in oven at 80 °C. The resulting slurry was calcinated at 250 °C for 2 h in the air to decompose nitrous completely. After grinding, the black oxide powders underwent calcination in air at 400 °C for 8 h. $Co_3O_4$ oxides were synthesized by the same method.

**Electrochemical characterizations**. The OER tests were operated in a three-electrode cell with a working electrode (WE) of glassy carbon flake (10 × 20 × 0.5 mm; Effective electrode area: 1.0 cm$^2$), a counter electrode of platinum foil, and a Hg/HgO reference electrode (RE) (filled with 1 M KOH solution). The catalysts electrode was fabricated by the recipe drop-castes method, which was reported in elsewhere[63]. The catalysts were mixed with acetylene black (AB) at a mass ration of 5:1, then were dispersed in isopropanol/water (v/v = 1:4) solvent followed by the addition of $Na^+$-exchanged Nafion as the binder. The mixtures were ultrasonicated for 30 min to reach homogeneous ink. The concentration of oxides in ink is 5 mg/ml, and AB is 1 mg/ml. Before drop-casting, the glassy carbon electrodes were polished to a mirror finish with $\alpha$-$Al_2O_3$ (50 nm) and washed by IPA and water to clean up completely. Finally, the as-prepared ink (100 ul) was dropped onto glassy carbon electrodes to reach a loading mass of 500 $\mu g_{ox}$ cm$^{-2}$ and the electrodes were dried overnight at room temperature. Cyclic voltammograms (CVs), linear sweep voltammetry (LSV), and chronoamperometry (CA) were performed in $O_2$-saturated 1 M KOH by using Bio-logic SP 150 potentiostat. All potentials were converted to RHE scale according to the following equation: RHE = Hg/HgO + 0.098 with iR correction. The tests of methanol oxidation reaction (MOR) and ethylene glycol oxidation reaction (EGOR) on CFO electrodes are similar to the OER test. The difference is that the MOR and EGOR were studied in 1 M KOH 100 ml electrolyte in the presence of 1 ml methanol and 1 ml ethylene glycol, respectively[64].

**Materials characterizations**. The X-ray diffraction (XRD) of oxides were carried on Bruker D8 diffractometer at a scanning rate of 2° min$^{-1}$, under Cu-K$_\alpha$ radiation ($\lambda$ = 1.5418 Å). DC magnetization measurements were performed on a Super-conducting Quantum Design (SQUID) magnetometer (MPMS-XL). The SQUID measurements of the magnetization of samples as a function of the magnetic field were carried out at 300 K in fields between −5 T and +5 T. The high-resolution transmission electron microscopy (HRTEM) was carried JEOL JEM- 2100 plus microscope at 200KV. The STEM results presented here were obtained using the 200 kV JEOL ARM electron microscope equipped (JEOL, Tokyo, Japan) with double aberration correctors, a dual-energy-loss spectrometer and a cold field emission source. The atomic-resolved STEM images were collected with a condense aperture of 28 mrad and a collection angle of 90–370 mrad for HAADF and 11–23 mrad for ABF images. The XPS measurements were performed using PHI-5400 equipment with Al Kα beam source (250 W) and a position-sensitive detector (PSD) was used to determine the surface composition of the materials. The Fourier transform infrared spectroscopy–Raman spectroscopy was carried with a confocal Raman microscope (Horiba HR Evolution), equipped with a diode laser emitting at 532 nm. The nominal laser power was filtered down to 1 mW to avoid sample overheating. Spectra were recorded with the accumulation time of 60 s.

**DFT studies**. All the density functional theory (DFT) calculations were performed by Vienna Ab-initio Simulation Package[65,66] (VASP), employing the Projected Augmented Wave[67] (PAW) method. The revised Perdew-Burke-Ernzerhof (RPBE) functional was used to describe the exchange and correlation effects.[68–70] The GGA + U calculations are performed using the model proposed by Dudarev et al.[71], with the $U_{eff}$ ($U_{eff}$ = Coulomb U – exchange J) values of 3.3 eV and 4 eV for Co and Fe, respectively. For all the geometry optimizations, the cutoff energy was set to be 500 eV. A 3 × 3 × 1 Monkhorst-Pack grids[72] was used to carry out the surface calculations on the (111) surface of $CoFe_2O_4$. At least 20 Å vacuum layer was applied in z-direction of the slab models, preventing the vertical interactions between slabs.

In alkaline conditions, OER could occur in the following four elementary steps:

$$OH^- + * \rightarrow *OH + e^- \tag{2}$$

$$OH^- + *OH \rightarrow *O + e^- \tag{3}$$

$$OH^- + *O \rightarrow *OOH + e^- \tag{4}$$

$$OH^- + *OOH \rightarrow * + O_2 + H_2O + e^- \tag{5}$$

where * denotes the active sites on the catalyst surface. Based on the above mechanism, the free energy of three intermediate states, *OH, *O, and *OOH, are important to identify a given material's OER activity. The computational hydrogen electrode (CHE) model[73] was used to calculate the free energies of OER, based on which the free energy of an adsorbed species is defined as

$$\triangle G_{ads} = \triangle E_{ads} + \triangle E_{ZPE} - T \triangle S_{ads} \tag{6}$$

where $\triangle E_{ads}$ is the electronic adsorption energy, $\triangle E_{ZPE}$ is the zero point energy

difference between adsorbed and gaseous species, and $T\triangle S_{ads}$ is the corresponding entropy difference between these two states. The electronic binding energy is referenced as ½ $H_2$ for each H atom, and ($H_2O$ – $H_2$) for each O atom, plus the energy of the clean slab. The corrections of zero point energy and entropy of the OER intermediates can be found in the Supplementary Table 2.

## Data availability

The data that support the findings of this study are available from the corresponding author upon reasonable request.

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

## Acknowledgements

We thank the support from the National Nature Science Foundation of China (Grant No. 11274370 and 51471185) and the National National Key R&D Program of China (Grant No. 2016YFJC020013 and 2018FYA0305800). Authors in Singapore thank the support from the Singapore Ministry of Education Tier 2 Grants (MOE2017-T2-1-009 and MOE2018-T2-2-027) and the Singapore National Research Foundation under its Campus for Research Excellence and Technological Enterprise (CREATE) programme. We appreciate the Facility for Analysis, Characterisation, Testing, and Simulation (FACTS) at Nanyang Technological University for materials characterizations. This work was supported by the National Research Foundation (NRF), Prime Minister's Office, Singapore under its Campus for Research Excellence and Technological Enterprise (CREATE) programme through the eCO2EP project operated by the Cambridge Centre for Advanced Research and Education in Singapore (CARES) and the Berkeley Educational Alliance for Research in Singapore (CARES).

## Author contributions

X.R., T.W., and Y.S. contribute equally to this work. Z.X., X.R., and T.W. conceived the original concept and designed the experiments. T.W. prepared the materials. X.R.

performed most characterizations and analysis. G.X. and X.R. performed the Raman spectroscopy measurement. H.Y., X.R., and Y.L. carried out magnetic property measurements and analysis. Y.S. contributed the DFT calculations and analysis. T.W., X.R., H.Y., H.G., and Z.X. contributed the mechanism analysis. J.G. contributed the explanation of QSEI theory. X.R., T.W., Y.S., H.Y., X.L., C.S., H.G., and Z.X. wrote the manuscript with the input from all authors. All authors engaged in the analysis of experimental results and manuscript edition.

## Competing interests

The authors declare no competing interests.
