## [Peer Review File · Nature Communications]

REVIEWER COMMENTS

Reviewer #1 (Remarks to the Author):

The manuscript describes experiments that verify the role of electron spin polarization in the oxygen evolution reactions. The work is timely and the experimental results are convincing. However the introduction provided is below the standard one would expect from any scientific publication. The paragraph are written in way that they cannot be understood. For example, the sone starting with: "It was recently notices...angular momentum." It combines scientific wrong statements. The issue of spin forbidden process in the oxygen evolution is known for decades and is well documented. In the sentence that follows the authors state that "..its next ground state (triplet states)." This sentence has no meaning.

In addition the authors do not cite theoretical works done long ago and that discuss this issue of spin forbidden reaction. See for example:

1. Chrétien, S.; Metiu, H. J. Chem. Phys. 2008, 129 (7), 74705.
2. Torun, E.; Fang, C. M.; de Wijs, G. A.; de Groot, R. A. J. Phys.Chem. C 2013, 117 (12), 6353–6357.

The work should be publish after major english editing and massive rewriting of the discussion.

Reviewer #2 (Remarks to the Author):

The authors of this manuscript studied the OER in alkaline media using ferromagnetic CoFe₂O₄ as catalyst. After applying a magnetic field to induce spin alignment the quantum spin-exchange interactions (QSEI) is suggested to facilitate the overall OER, by favoring the formation of O₂ in the more stable triplet state. Non-ferromagnetic reference catalysts, Co₃O₄ and IrO₂, do not show an effect on their OER activity in when a magnetic field is applied. Overall, the work is rather similar to that in Ref [29] and provides no notable new insight into the underlying phenomena (Garcés-Pineda, F.A., Blasco-Ahicart, M., Nieto-Castro, D., López, N. & Galán-Mascarós, J.R. Direct magnetic enhancement of electrocatalytic water oxidation in alkaline media. Nature 480 Energy 4, 519-525 (2019)). In particular, the theoretical work has major deficiencies and I cannot recommend the article for publication in Nature Communications.

The authors make Tafel slope arguments that the "first" reaction step, i.e., the adsorption of OH⁻ is rate-limiting. In contrast, the spin argument relates to the formation of O₂ in its triplet state. OH⁻ is a non-magnet singlet reactant and it adsorbs as OH*^{*}; it is unclear why a magnetic field would make this step faster and shift the rate-determining step to between 1 and 2.

Frankly, it does not even make sense to define a "first" reaction in a catalytic cycle. One can easily argue that the "first" step is O* + OH⁻  *OOH + e⁻. Here, O* could be a lattice oxygen on the spinel surface or an oxygen adatom on a (partially) oxidized spinel surface (see next comment suggesting a Pourbaix diagram). If this was the "first" step, it might actually make sense why a magnetic field can favor the triplet state in the intermediate *OOH species, which forms the O-O bond for O₂.

Which surface termination of the (111) facet was used? Stoichiometric, oxidized or reduced? A surface Pourbaix diagram would be helpful to determine the likely surface termination under reaction conditions.

No barriers were calculated for OH⁻ addition. There is some evidence that proton + electron addition during ORR in acidic medium is fast, but can the kinetics of OH⁻ addition also be assumed to be fast? A step of particular concern is O* + OH⁻  OOH* + e⁻. This step forms an O-O bond and a significant activation barrier can be expected.

The description of the DFT methods is insufficient. A key conclusion that is drawn relates to the magnetic moments/spin, but no spin-related information is provided. What magnetic structures of the CoFe₂O₄ (and Co₃O₄ or IrO₂) were found? What were the initial guesses and the final magnetic moments? For AFM structures, which atoms had the up, which had the down spin? Is there a difference if a tetrahedral vs. an octahedral atom has the up/down spin?

Where do the intermediates adsorb? To tetrahedral or octahedral sites?

Spinel can have any degree of inversion. Was the CoFe₂O₄ modeled as normal spinel, inverse spinel, partially inverted spinel? To what extent has the degree of inversion an effect on the calculated values?

Without compelling evidence for the proposed spin-dependent kinetics of the OER, this manuscript simply lacks the novelty necessary for publication in Nature Communications.

Reviewer #3 (Remarks to the Author):

General comment:

The present work deals with the oxygen evolution reaction (OER) enhancement using constant magnetic field and ferromagnetic ordered catalysts. This study exhibits spin selection as a possible way to promote OER in alkaline conditions. The authors achieved rigorous methodologies to reveal that spin polarization occurs at the first electron transfer step in OER. This work will stand out in the understanding and design of spin-dependent catalysts.

Major comments:

Many comments are to be discussed because "Lorentzian movement" is not limited only to the magnetic field and the velocity field. Mostly, Lorentz force is expressed following equation:

$$j = \sigma B (E + u \times B)$$

Where, the Gradient of potential E , the flow u , the magnetic field B and the conductivity of liquid phase σ . However, magnetic effect involved in the momentum balance is more complex, e.g. the Kelvin Force should be considered and other effects due to the gradient of magnetic field:

$$u = - \frac{\chi_{mag}}{\mu_{mag}} \nabla B^2$$

χ_{mag} is the mass magnetic susceptibility of oxygen, μ_{mag} the magnetic permeability (for: water and oxygen mixture), and B is the magnetic flux density. Therefore, "Lorentzian movement" and consecutive mass transport process depends on the local gradients of magnetic field [1]. In addition, the sign of the charge adsorbed by the bubble, affects the mass transport [2]

Page 6 lines 132 to 134: the affirmation of a weak effect on mass transport should be discussed more in depth. Figure 5b: CA measurements under the different magnetic field strength in O₂-saturated 1 M KOH is not clear. HRTEM images of Co₃O₄, and IrO₂ catalysts should be provided in supplementary material. Page 14 Lines 278: it is not correct; the gradient of magnetic field was not investigated. Only the magnetic field strength effect was considered.

According to D. W. Banham et al. [3], Tafel slope depends on microstructure of catalyst layer and electrolyte conductivity. Consequently, the change of Tafel slope could appear for a different set of operating conditions. Tafel plots of CoFe₂O₄, Co₃O₄, and IrO₂ catalysts with various temperature is required to consolidate discussion page 10 lines 202 to 209. Microstructure characterizations of each catalytic layer are also required.

Page 7 a subsection entitled: "No surface restructuring in OER was developed". Of course spinel crystal structure of CoFe₂O₄ remained after the electrochemical treatment + Raman + X-ray

photoelectron spectroscopy. However, to evidence a non-modification surface, HRTEM should be performed at exactly same location, but the Figure S4 showed two different places. Please improve this critical point in this section.

- [1] V. Gatard et al., « Use of magnetic fields in electrochemistry: A selected review », *Curr. Opin. Electrochem.*, vol. 23, p. 96-105, oct. 2020, doi: 10.1016/j.coelec.2020.04.012.
- [2] L. M. A. Monzon et al., « Magnetically-Induced Flow during Electropolishing », *J. Electrochem. Soc.*, vol. 165, no 13, p. E679-E684, 2018, doi: 10.1149/2.0581813jes.
- [3] D. W. Banham et al. « Pt/Carbon Catalyst Layer Microstructural Effects on Measured and Predicted Tafel Slopes for the Oxygen Reduction Reaction », *J. Phys. Chem. C*, vol. 113, no 23, p. 10103-10111, juin 2009, doi: 10.1021/jp809987g.

Response to the comments of Reviewers

We would like to thank the reviewers for giving us comments as well as valuable suggestions to our manuscript. We have revised the manuscript according to the reviewers' comments and all the changes are highlighted in red color in the revised manuscript. Below please find a point-to-point response to the reviewers' comments.

Reviewer #1

The manuscript describes experiments that verify the role of electron spin polarization in the oxygen evolution reactions. The work is timely and the experimental results are convincing. However the introduction provided is below the standard one would expect from any scientific publication. The paragraphs are written in a way that they cannot be understood. For example, the sentence starting with: "It was recently noticed...angular momentum." It combines scientific wrong statements. The issue of spin forbidden process in the oxygen evolution is known for decades and is well documented. In the sentence that follows the authors state that ". its next ground state (triplet states)." This sentence has no meaning.

In addition the authors do not cite theoretical works done long ago and that discuss this issue of spin forbidden reaction. See for example:

1. Chrétien, S.; Metiu, H. J. Chem. Phys. 2008, 129 (7), 74705.
2. Torun, E.; Fang, C. M.; de Wijs, G. A.; de Groot, R. A. J. Phys. Chem. C 2013, 117 (12), 6353–6357.

The work should be published after major English editing and massive rewriting of the discussion.

Response: We appreciate the reviewer for his/her positive comments and valuable suggestions. We have rephrased these sentences and cited those theoretical works in revised manuscript as references 17 and 18. Please see below for updated version:

Changes:

1). Line 50-54, Page 3:

"Besides, the produced O_2 is in triplet ground state, where the frontier π^* orbitals are occupied by two electrons with parallel alignment. In contrast, the ground spin state of reactant OH/H_2O is singlet with all paired electrons. In theory, the rate of a chemical reaction will be slow if the spin of the electronic wave function of the products differs from those of the reactants, as the Hamiltonian does not contain spin operators.^{17,18}"

2). Line 54-56, Page 3:

"The singlet states of the oxygen molecule were reported at an energy level of at least ~ 1 eV higher than its triplet state.^{18, 19}"

Reviewer #2

The authors of this manuscript studied the OER in alkaline media using ferromagnetic CoFe_2O_4 as catalyst. After applying a magnetic field to induce spin alignment the quantum spin-exchange interactions (QSEI) is suggested to facilitate the overall OER, by favoring the formation of O_2 in the more stable triplet state. Non-ferromagnetic reference catalysts, Co_3O_4 and IrO_2 , do not show an effect on their OER activity in when a magnetic field is applied. Overall, the work is rather similar to that in Ref [29] and provides no notable new insight into the underlying phenomena (Garcés-Pineda, F.A., Blasco-Ahicart, M., Nieto-Castro, D., López, N. & Galán-Mascarós, J.R. Direct magnetic enhancement of electrocatalytic water oxidation in alkaline media. *Nature Energy* 4, 519-525 (2019)). In particular, the theoretical work has major deficiencies and I cannot recommend the article for publication in *Nature Communications*.

Response: Thanks for the reviewer's comments. The reference [29] (*Nature Energy* 2019, 4, 519-525) opens opportunities for implementing magnetic enhancement in water splitting. Inspired by this reference, we tried to understand more on how the magnetic field-induced spin polarization affects the OER process. Our work is different from Ref [29]. The main novelties of our work are outlined below, which are not present in Ref [29]:

1. We have found that ferromagnetic CoFe_2O_4 as the spin polarizer facilitates the spin polarization under a constant magnetic field because macroscopic ferromagnetic QSEI promote the OER activity. The increase of spatial spin polarization shows a positive correlation with the enhancement of spin-transport (selection) during OER. We have found that the Tafel slope of overall ferromagnetic CoFe_2O_4 switched from ~ 120 to ~ 90 $\text{mV}\cdot\text{dec}^{-1}$ after applying a magnetic field. It indicates the change of the rate-determining step (RDS) of OER reaction under an external magnetic field, i.e., the first electron transfer step is no longer the RDS. The spin-polarized electron exchange between the ferromagnetic CoFe_2O_4 and the adsorbed oxygen species (reactants) for the first electron transfer is ferromagnetic exchange with the lower the electron-electron repulsion, which leads to faster reaction kinetics for the first electron transfer step.
2. We have conducted a DFT study on spin density of CoFe_2O_4 with and without spin alignment. Compared to the CoFe_2O_4 with anti-parallel couplings, the CoFe_2O_4 with aligned spin has a higher spin density on the oxygen atoms and an increment of the 3d-2p hybridization. Such changes in electronic structure are associated with ferromagnetic ligand holes near the Fermi level, which can thereby facilitate spin-polarized charge transport and optimize the kinetics of the spin-charge transfer. We also conducted a DFT study on the free energies of OER steps on the (111) surface of CoFe_2O_4 with and without spin alignment. The active sites with aligned spin are thermodynamically more favorable to OER and the overpotential of producing triplet oxygen is reduced by 390 mV compared to that without aligning spin at the first electron transfer step.
3. We further investigate the reaction involving non-magnetic molecules on CoFe_2O_4 under the magnetic field, such as the methanol oxidation reaction (MOR) and

ethylene glycol oxidation reaction (EGOR). We have found that the magnetic field has negligible effect on the reaction kinetics.

4. We also performed the measurement under the different strength of the magnetic field. The increment is positively correlated with the magnetization of the ferromagnetic CoFe_2O_4 . Such an investigation provides the direct evidence to the spin-polarization effect on OER enhancement.

The authors make Tafel slope arguments that the “first” reaction step, i.e., the adsorption of OH^- is rate-limiting. In contrast, the spin argument relates to the formation of O_2 in its triplet state. OH^- is a non-magnet singlet reactant and it adsorbs as OH^* ; it is unclear why a magnetic field would make this step faster and shift the rate-determining step to between 1 and 2.

Response: Thanks for the reviewer’s comments. As Tafel plots of CoFe_2O_4 shown, the Tafel slope of CoFe_2O_4 is about $109 \text{ mV}\cdot\text{dec}^{-1}$ and that indicates the first electron transfer from the adsorbed OH^- is rate-limiting. Similar results have been reported in literature, e.g. *Materials Chemistry and Physics*, 2019, 237, 121847; *ACS Appl. Energy Mater.*, 2019, 2, 1026–1032.

As the reviewer stated, the formation of triplet state O_2 relates to violate conservation of angular momentum from a non-magnet singlet reactant (OH^-). In this work, we found that ferromagnetic CoFe_2O_4 as the spin polarizer facilitates the spin polarization under a constant magnetic field. The electron transfer at the catalytic interface depends on the transition probability, which is associated with the wavefunction integral between OH^- and the active site. When the electron of the adsorbed oxygen species (OH^-) transfer to the ferromagnetic catalyst, the ferromagnetic exchange will happen more easily due to the low electron-electron repulsion (show in Figure R1b & Figure 4b). As a result, the first electron transfer process led to the generation of $\text{O}(\downarrow)^-$, that is, the first electron transfer step is spin polarization process. In the next electron transfer step, the fresh OH^- will loss an electron with the fixed spin direction to form the triplet state intermediate $\text{O}(\downarrow)\text{O}(\downarrow)\text{H}$ species with a lower barrier (as shown in Figure R2 & Supplementary Figure 7). Consequently, the triplet state intermediate $\text{O}(\downarrow)\text{O}(\downarrow)\text{H}$ species will prefer to generate the triplet state O_2 . The DFT calculation of free energies of OER steps on CoFe_2O_4 also show that the triplet state in the intermediate $\text{O}(\downarrow)\text{O}(\downarrow)\text{H}$ species transfer to triplet state O_2 is more thermodynamically favorable. Thus, with $\text{O}(\downarrow)^-$ in first electron transfer step, the spin polarization has been facilitated before the O-O coupling.

Figure R1 & Figure 4 (a) Schematic of spin-exchange mechanism for OER. **(b)** The first electron transfer step is promoted by spin polarization through the FM exchange (QSEI), which gives smaller electronic repulsions and makes the adsorbed O species have a fixed spin direction. The corresponding inter-atomic wavefunction is shown in the right of this panel. **(c)** The spin density of CoFe_2O_4 with and without aligned spin. **(d)** The free energy diagram of OER at 1.23 V (vs. RHE) with and without the aligned spin on the (111) surface of CoFe_2O_4 toward triplet oxygen production.

Figure R2 & Supplementary Figure 7. The production of the triplet intermediate $O(\downarrow)O(\downarrow)H$ species.

To further improve the manuscript, Figure 4a have been modified in the revised manuscript and Figure R2 has been added in the revised SI as Supplementary Figure 7. Related discussion has been added in the revised manuscript. (Line 258, Page 12, marked in red):

“As a result, the first electron transfer process led to the generation of $O(\downarrow)^-$, that is, the first electron transfer step is spin polarization process. In the next electron transfer step, the fresh OH^- will lose an electron with the fixed spin direction to form the triplet state intermediate $O(\downarrow)O(\downarrow)H$ species with a lower barrier (Supplementary Figure 7). Consequently, the triplet state intermediate $O(\downarrow)O(\downarrow)H$ species will prefer to generate the triplet state O_2 .”

Frankly, it does not even make sense to define a “first” reaction in a catalytic cycle. One can easily argue that the “first” step is $O^* + OH^- \rightarrow ^*OOH + e^-$. Here, O^* could be a lattice oxygen on the spinel surface or an oxygen adatom on a (partially) oxidized spinel surface (see next comment suggesting a Pourbaix diagram). If this was the “first” step, it might actually make sense why a magnetic field can favor the triplet state in the intermediate *OOH species, which forms the O-O bond for O_2 .

Response: We appreciate the reviewer for insightful opinions on the reaction mechanisms regarding the spin polarization in OER. We totally agree with that defining a “first” reaction is not rigorous in a catalytic cycle. We want to clarify that the “first” here should specifically refer to the first electron transfer step from $^*OH^-$ in an OER cycle. In OER process, the electron transfers firstly from a metal center with the deprotonation of OH^- . When the catalyst is ferromagnetic ordering, the orbitals of the ferromagnetic catalyst create an intrinsically degenerate spin-polarized metallic state that optimizes the wavefunction based on the inter-atomic reduction of the electron-electron repulsion. Thus, the spin polarization facilitate the first electron transfer step (Figure R3a). In the next reaction step, the OH^- will lose an electron with the fixed spin direction to form the triplet state

intermediate $O(\downarrow)O(\downarrow)H$ species with a lower barrier (as shown in Figure R2). Then, the triplet state intermediate $O(\downarrow)O(\downarrow)H$ species give the triplet state O_2 .

We address that the reaction started between the oxygen atom on the surface (i.e. a lattice oxygen on the oxide surface or an oxygen adatom on a partially oxidized spinel surface, as commented by the reviewer) and the adsorbed oxygen species (OH^\cdot), the “first” step is $O^* + OH^- \rightarrow ^*OOH + e^-$. In the ferromagnetic ordering catalyst, the oxygen atoms and a concomitant increment of the 3d-2p hybridization, which associate with ferromagnetic ligand holes, facilitate the spin-selected charge transfer to form the triplet state intermediate $O(\downarrow)O(\downarrow)H$ species (shown in Figure R3b). The first electron transfer from the adsorbed OH^\cdot is spin-direction selected due to the spin polarized oxygen on the oxide surface. It further promotes the step-by-step generation of the triplet state O_2 . It should be noted that the free energy of $O^* + OH^- \rightarrow ^*OOH + e^-$ is not downhill as shown in Figure 4d. Thus, we believe that the electron will transfer firstly from the adsorbed oxygen species (OH^\cdot) to the metal center.

Overall, in the OER process on a ferromagnetic ordering catalyst, we conclude that the spin polarization takes place in the first electron transfer step of OER.

Figure R3. The spin polarization mechanisms in OER: (a) on the metal sites, (b) on the oxygen sites.

To further improve the manuscript, all the “first step” in the manuscript have been revised as “the first electron transfer step”.

Which surface termination of the (111) facet was used? Stoichiometric, oxidized or reduced? A surface Pourbaix diagram would be helpful to determine the likely surface termination under reaction conditions.

Response: Thanks for the comments. We noted that the surface termination of the (111) facet have two surface planes in spinel oxides (Adv. Mater. 2019, 31, 1902509) as displayed in Figure R4. In this work, the cobalt atoms have taken as active site which mainly distributed in octahedral sites. Some experimental and computational data also show that the preferential exposure of octahedral site is a more general property of spinel

oxides (Adv. Mater. 2019, 31, 1902509; *J. Phys. Chem. C* 2016, 120, 19087). Thus, the exposure of occupied octahedral sites has been used in this DFT work.

Figure R4. (111) surface planes of spinel oxide.

No barriers were calculated for OH⁻ addition. There is some evidence that proton + electron addition during ORR in acidic medium is fast, but can the kinetics of OH⁻ addition also be assumed to be fast? A step of particular concern is O* + OH⁻ → OOH* + e⁻. This step forms an O-O bond and a significant activation barrier can be expected.

Response: This is a great viewpoint. We would like to explain why no barriers were calculated. In this work, the aim of performing DFT calculations is to explore how the OER free energy diagram of CoFe₂O₄ evolves under an applied magnetic field. The computational hydrogen electrode (CHE) model proposed by Norskov et al. (JPCB 2004, 108, 17886-17892) was used to perform the OER free energy calculations. In this model, no barriers are involved and the rate-determining step (RDS) is considered as the elementary step with the largest uphill free energy change. The rationality of this approach is that the kinetic barrier of two reaction states is proportional to the free energy gap of them, which is governed by the Bell–Evans–Polanyi (BEP) relation (Figure R5). Therefore, to identify the RDS, no barriers were needed to calculate. Besides, based on the definition of the four elementary steps for OER calculations,

OH⁻ addition is involved in each of the four steps. Thus, the relative magnitudes of the kinetics for each OH⁻ addition can also be determined by evaluating the free energy changes between two reaction states. And in this calculation, since the first elementary step shows the highest uphill free energy change, we deduce that this step should be the RDS and with the highest activation barrier.

Figure R5. The Bell–Evans–Polanyi (BEP) relation (Phys. Chem. Chem. Phys., 2005, 7, 2552–2553).

The description of the DFT methods is insufficient. A key conclusion that is drawn relates to the magnetic moments/spin, but no spin-related information is provided. What magnetic structures of the CoFe_2O_4 (and Co_3O_4 or IrO_2) were found? What were the initial guesses and the final magnetic moments? For AFM structures, which atoms had the up, which had the down spin? Is there a difference if a tetrahedral vs. an octahedral atom has the up/down spin?

Response: Thanks for the suggestions. We have added the description of the DFT methods in the Supplementary Information. Specifically, for the computational details, we would like to give some more details of the OER calculations under a magnetic field. The function of the outer magnetic field is to align all the randomly oriented spin in the catalyst to a specific direction. To model this situation, we have used the ‘LNONCOLLINEAR’ and ‘SAXIS’ keywords to make the spin in the catalyst to a specific direction. And during the calculations, we did not set initial guesses of the magnetic moments and let VASP to fully relax until finding out the most stable configuration. In the following table, we have summarized the final magnetic moments (μ_B) of the metal cations after structural optimization. For the reviewer’s question regarding the AFM configurations, we are confused as we did not perform the calculations of any AFM configuration in this work.

Table R1 & Supplementary Table 3. The magnetic moments (μ_B) of the metal cations before and after structural optimization.

	Fe (Tet)	Fe (Oct)	Co (Oct)
Without aligned CoFe_2O_4	4.224	4.087	0.869
With aligned CoFe_2O_4	2.451	2.487	0.530

Table R1 have been added into the revised SI as Supplementary Table 3. The related discussion has been added in the revised manuscript (Line 95, Page 4, marked in red):

“The details of computational under magnetic field are as follows. The function of the outer magnetic field is to align all the randomly oriented spin in the catalyst to a specific direction. To model this situation, we have used the ‘LNONCOLLINEAR’ and ‘SAXIS’ keywords to make the spin in the catalyst to a specific direction. And during the calculations, we did not set initial guesses of the magnetic moments and let VASP to fully relax until finding out the most stable configuration. In the Supplementary Table 3, we have summarized the final magnetic moments (μ_B) of the metal cations after structural optimization.”

Where do the intermediates adsorb? To tetrahedral or octahedral sites?

Response: Thanks for the questions. In this work, the octahedral sites of CoFe_2O_4 have been used as active sites. We address below the concern on the intermediates adsorb. First, CoFe_2O_4 spinel structure is predominantly inverse with Co^{2+} ions mainly in octahedral sites and Fe^{3+} ions almost equally in octahedral and tetrahedral sites. The octahedral and tetrahedral sites are antiferromagnetically coupled, resulting in a ferrimagnetic property of the compound. The ferrimagnetic moments come mostly from Co^{2+} in octahedral site. The theoretical calculation show $M_{\text{Fe}^{3+}} = 5 \mu_B$ and $M_{\text{Co}^{2+}} = 3.5 \mu_B$ (Journal of Magnetism and Magnetic Materials, 1993, 123, 93-96).

Here, we have added additional data to confirm an inversed spinel structure of CoFe_2O_4 . The temperature-dependent magnetizations were measured with a magnetic field (H) of 100 Oe under field-cooling procedures for CoFe_2O_4 (Figure R6). In the high temperature area, the susceptibilities derived from the magnetizations ($\chi = M/H$) obey a Curie–Weiss law: $\chi = C/T - T_C$, where C is Curie constant, and T_C is Curie–Weiss temperature. By fitting the susceptibility versus. T data in Figure R6, an effective magnetic moment μ_{eff} can be obtained through $\mu_{eff} = \sqrt{8C} \mu_B$. Here, the calculated μ_{eff} of $3.44 \mu_B$ for the CoFe_2O_4 sample is very close to the idea inverse spinel value. Thus, the used CoFe_2O_4 catalysts is almost in a perfect inverse spinel structure. The Fe^{3+} ions distributed almost equally in octahedral and tetrahedral sites with the spin in opposite direction and Co^{2+} ions mainly distributed in octahedral sites which contribute effective magnetic moment. Furthermore, extended X-ray absorption fine structure (EXAFS) fitting of the characteristic octahedra peak and tetrahedra peak gives the cation distribution. It can be seen that CoFe_2O_4 cubic

spinels, $\approx 90\%$ Co cations occupy octahedral (Figure R7). Thus, we consider the Co cation as active sites, which mainly stay in octahedral sites.

Figure R6. The field-cooled M-T curves of CoFe_2O_4 . The inset figure shows the temperature dependence of reciprocal susceptibilities. The solid line is the fitting results by the Curie–Weiss law.

	Tetrahedral site	Octahedral site
Co _x	0.109	0.891
Co-O (Å)	1.81	2.06
Co-O coordination No.	3.3	4.9
Fe _y	0.891	1.109
Fe-O (Å)	1.93	1.99
Fe-O coordination No.	3.0	4.5
$\chi^2_{\text{red}} = 376.9$; R factor = 0.0188		

Figure R7. EXAFS $k^3\chi(R)$ spectra (gray circles) and fitting results (solid lines) of CoFe_2O_4 oxides at Co and Fe K-edge. The table show the summary of EXAFS fitting results for CoFe_2O_4 .

Spinels can have any degree of inversion. Was the CoFe_2O_4 modeled as normal spinel, inverse spinel, partially inverted spinel? To what extent has the degree of inversion an effect on the calculated values?

Response: Thanks for the questions. As replied above, the CoFe_2O_4 employed here is almost in a perfect inversion spinel structure (characterized by magnetic and EXAFS tests). Thus, the CoFe_2O_4 was modeled as an inverse spinel for calculation.

Without compelling evidence for the proposed spin-dependent kinetics of the OER, this manuscript simply lacks the novelty necessary for publication in Nature Communications.

Reviewer #3

General comment:

The present work deals with the oxygen evolution reaction (OER) enhancement using constant magnetic field and ferromagnetic ordered catalysts. This study exhibits spin selection as a possible way to promote OER in alkaline conditions. The authors achieved rigorous methodologies to reveal that spin polarization occurs at the first electron transfer step in OER. This work will stand out in the understanding and design of spin-dependent catalysts.

Major comments:

Many comments are to be discussed because “Lorentzian movement” is not limited only to the magnetic field and the velocity field. Mostly, Lorentz force is expressed following equation:

$$j = \sigma B (E + u \times B)$$

Where, the Gradient of potential E , the flow u , the magnetic field B and the conductivity of liquid phase σ . However, magnetic effect involved in the momentum balance is more complex, e.g. the Kelvin Force should be considered and other effects due to the gradient of magnetic field:

$$u = -\frac{\chi_{mag}}{\mu_{mag}} \nabla B^2$$

χ_{mag} is the mass magnetic susceptibility of oxygen, μ_{mag} the magnetic permeability (for: water and oxygen mixture), and B is the magnetic flux density. Therefore, “Lorentzian movement” and consecutive mass transport process depends on the local gradients of magnetic field [1]. In addition, the sign of the charge adsorbed by the bubble, affects the mass transport [2]

Page 6 lines 132 to 134: the affirmation of a weak effect on mass transport should be discussed more in depth. Figure 5b: CA measurements under the different magnetic field strength in O₂-saturated 1 M KOH is not clear. HRTEM images of Co₃O₄, and IrO₂ catalysts should be provided in supplementary material. Page 14 Lines 278: it is not correct; the gradient of magnetic field was not investigated. Only the magnetic field strength effect was considered.

Response: Thanks for the suggestions. We understand the concern from reviewer about the effect on mass transport under the magnetic field. The Lorentz force can affect the charged ions moving in electrolyte, which leads to consecutive mass transport process (Current Opinion in Electrochemistry 2020, 23, 96–105) The Lorentz force decreases the thickness of the diffusion layer, which thus increases the limiting current density of a cathodic reaction, such as metal electrodeposition (*Electrochimica Acta*. 2007, 53, 161-166.). However, it is very different for OER in aqueous solution. It has been well

understood that OH^- and H_3O^+ in aqueous solution do not move physically, but by sequential proton transfer, known as Grotthuss mechanisms (Figure R8a and 8b). That means the influence of Lorentz force on the physical movement of ions OH^- or H_3O^+ is negligible. On the other hand, there are reports years ago that the magnetohydrodynamic effect (MHD) which originates from the Lorentz force can improve the release of gas bubbles in HER and OER and thus to improve the reaction kinetics (*Electrochimica Acta*, 2013, 100, 261-264; *Journal of The Electrochemical Society*, 2018, 165, E679-E684). However, it should be noticed that those reports are for the region of high overpotential and high current, in which region the release of bubbles significantly affect the reaction kinetics. This is not the case in our experiments. Moreover, if the MHD effect or other effects (e.g. ref electrode, counter electrode, and resistivity) do have influence on the OER activity under magnetic field, similar enhancement should also be observed for methanol oxidation reaction (MOR) and ethylene glycol oxidation reaction (EGOR). However, it is not the case. The activity enhancement is uniquely found for OER that involves spin-related electron transfer.

Figure R8 & Supplementary Figure 3. The mechanism of proton hopping (jump) for (a) H_3O^+ and (b) OH^- in aqueous solution (*JPCL*, 2014, 5, 2568-2572.).

To further improve the manuscript, Figure R8 has been added in the revised SI as Supplementary Figure 3. Related discussion is added in the revised manuscript (Line 132, Page 6 marked in red):

“It also should be noted that OH^- and H_3O^+ in aqueous solution do not move physically, but by sequential proton transfer, known as Grotthuss mechanisms (Supplementary Figure 3). That means the influence of Lorentz force on the physical movement of ions OH^- or H_3O^+ is negligible.^{37”}

Thanks for the comments about the CA measurements. The modifications (as shown in Figure 5b) have been updated in Figure 6b in the revised manuscript and added the detail of CA measurements in Supplementary Information (Line 44, Page S2, marked in red)

“CA test under the different magnetic field strength (0, 500, 1000, 3000, 5000, 7500, and 10000 Oe) at a constant potential of 1.66 V versus RHE for CoFe₂O₄, Co₃O₄, and 1.56 V versus RHE for IrO₂.”

Figure R9 & Figure 6 | The effect of gradient magnetic field, remanence, and demagnetization. (a) Initial magnetization curve of CoFe₂O₄. (b) CA test in 1 M KOH under the different magnetic field strength (0, 500, 1000, 3000, 5000, 7500, and 10000 Oe) at a constant potential of 1.66 V versus RHE for CoFe₂O₄, Co₃O₄, and 1.56 V versus RHE for IrO₂. (c) The increment of the current density under different magnetic field strength. It was calculated by the following equation: Increment (%) = $(j_M - j_{M=0}) / j_{M=0}$; j_M is the chronopotentiometry current density values obtained under the applied magnetic fields (0, 500, 1000, 3000, 5000, 7500, and 10000 Oe). The error bar represents three independent tests. (d) LSV curves of CoFe₂O₄ at a scan rate of 10 mV/s in O₂-saturated 1 M KOH with and without a constant magnetic field (10000 Oe), after the magnetic field removed (after M), and after demagnetization. The corresponding Tafel plots are shown

in (f). (e) The magnetization of CoFe_2O_4 after removing a constant magnetic field of 10000 Oe. (g) The curve of demagnetization for CoFe_2O_4 .

We added the HRTEM images of Co_3O_4 , and IrO_2 catalysts in Figure R10 and Supplementary Figure 5.

Figure R10 & Supplementary Figure 5. HRTEM images of CoFe_2O_4 before (a) and after OER (b). HRTEM images of Co_3O_4 before (c) and after OER (d). HRTEM images of IrO_2 before (e) and after OER (f).

We have corrected “the gradient of magnetic field” as “the magnetic field strength” in the revised manuscript. There is no gradient of magnetic field as we applied constant magnetic field in the experiment using an electromagnet. Please see our set up in Figure R11.

Figure R11. The setup of OER test under the magnetic field by an electromagnet.

According to D. W. Banham et al. [3], Tafel slope depends on microstructure of catalyst layer and electrolyte conductivity. Consequently, the change of Tafel slope could appear for a different set of operating conditions. Tafel plots of CoFe_2O_4 , Co_3O_4 , and IrO_2 catalysts with various temperature is required to consolidate discussion page 10 lines 202 to 209. Microstructure characterizations of each catalytic layer are also required.

Response: Thanks for the comments. We totally agree with the reviewer that the Tafel slope depends on microstructure of catalyst layer and electrolyte conductivity. In this work, it should be noted that we compare the Tafel slope of catalysts with and without magnetic field in the same electrode. The microstructure of catalyst layer and electrolyte conductivity do not change. As requested by the reviewer, we added the SEM images of each catalyst. As shown in Figure R12, the Microstructure of CoFe_2O_4 , Co_3O_4 , and IrO_2 before and after OER under magnetic field have no remarkable difference.

Figure R12 & Supplementary Figure 10. SEM images of CoFe₂O₄ before (a) and after OER (b); SEM images of Co₃O₄ before (c) and after OER (d); SEM images of IrO₂ before (e) and after OER (f).

To further improve the manuscript, the Figure R12 have been added in the revised SI as Supplementary Figure 10. The related discussion has been added in the revised SI (Line 168, Page S11, marked in red).

“As shown in Supplementary Figure 10, the microstructure of CoFe₂O₄, Co₃O₄, and IrO₂ before and after OER test under magnetic field have no remarkable difference observed.”

According to the comments of the reviewer, we have carried out measurements of CoFe_2O_4 , Co_3O_4 , and IrO_2 catalysts under different temperatures as shown in Figure R13a. We first noted that the OER performance of catalysts is getting better as the reaction temperature increases. This is because that the rate constant for reaction will increase as the reaction temperature increases, which can promote this reaction based on the transition state theory (ACS Catal. 2020, 10, 4160–4170). More importantly, the OER performance of the ferromagnetic CoFe_2O_4 is promoted obviously under the magnetic field at room temperature (RT: ~ 303 K). As the reaction temperature increases, magnetic field enhancement OER performance of the ferromagnetic CoFe_2O_4 can still be clearly observed. The corresponding Tafel plots are shown in Figure R13b. At room temperature, the Tafel slope of CoFe_2O_4 is about $106 \text{ mV}\cdot\text{dec}^{-1}$ without the magnetic field. After applying a constant field, the Tafel slope decreases to circa $82.8 \text{ mV}\cdot\text{dec}^{-1}$. As the temperature increases, the magnetic field will still cause the Tafel slope of CoFe_2O_4 decreases. But the effect of magnetic field become not that remarkable under high temperature. This is because the arrangement of magnetic moments of catalyst will be thermally disturbed. The ferromagnetic ordering in the catalyst gets disturbed and thus a certain degree of demagnetization at high temperature occurs, which lead to the decreased influence of the magnetic field on OER.

Figure R13 & Figure 3. (a) LSV curves of CoFe_2O_4 at a scan rate of 10 mV/s in O_2 -saturated 1 M KOH with and without a constant magnetic field (10000 Oe) under the different temperatures (room temperature (RT): ~ 303 K, 308K, 318K and 328 K). The corresponding Tafel plots are shown in (b). (c) LSV curves of Co_3O_4 with and without a constant magnetic field (10000 Oe) under the different temperatures (room temperature (RT): ~ 303 K, 318K and 323 K). The corresponding Tafel plots are shown in (d). (e) LSV curves of IrO_2 with and without a constant magnetic field (10000 Oe) under the different temperatures (room temperature (RT): ~ 303 K, 318K and 323 K). The corresponding Tafel plots are shown in (f). Tafel slopes at various temperatures are summarized in (g).

To further improve the manuscript, Figure R13 have been added in the revised manuscript as Figure 3. Related discussion is added in the revised manuscript (Line 209, Page 10, marked in red):

“Furthermore, we have carried out OER measurements of CoFe_2O_4 , Co_3O_4 , and IrO_2 under different temperatures as shown in Figure 3a. We first noted that the OER performance of catalysts is getting better as the reaction temperature increases. This is because that the rate constant for reaction will increase as the reaction temperature increases, which can promote this reaction based on the transition state theory.⁵² More importantly, the OER performance of the ferromagnetic CoFe_2O_4 is promoted under the magnetic field at various temperature. However, the positive influence of the magnetic field on the OER performance of CoFe_2O_4 is decreased as the reaction temperature increases. The corresponding Tafel slopes are shown in Figure 3b. At room temperature, the Tafel slope of CoFe_2O_4 is about $106 \text{ mV}\cdot\text{dec}^{-1}$ without the magnetic field. After applying a constant field, the Tafel slope decreases to circa $82.8 \text{ mV}\cdot\text{dec}^{-1}$. As the temperature increases, the positive influence of the magnetic field became not that remarkable. This is because the arrangement of magnetic moments of catalyst will be thermally disturbed. The ferromagnetic ordering in the catalyst gets disturbed and thus a certain degree of demagnetization at high temperature occurs, which lead to the decreased influence of the magnetic field on OER.”

Page 7 a subsection entitled: “No surface restructuring in OER was developed”. Of courses spinel crystal structure of CoFe_2O_4 remained after the electrochemical treatment + Raman + X-ray photoelectron spectroscopy. However, to evidence a non-modification surface, HRTEM should be performed at exactly same location, but the Figure S4 showed two different places. Please improve this critical point in this section.

Response: Thanks for the comments. According to the comments of the reviewer, we have carried out HRTEM observation on the CoFe_2O_4 catalyst at identical location before and after electrochemical treatments using identical location TEM technique (IL-TEM).

A diluted ink containing catalyst was pipetted onto the gold finder grid (400 mesh, TED PELLA, USA). The pristine catalyst on the gold finder grid was observed and mark the specific location before cycling. Then the grid was used at the working electrode in an electrochemical cell. After the electrochemical cycling, the grid was dried under Ar flow and observed under TEM again. The specific location on the grid helped to find the particle at the same location as it is before electrochemical cycling. The images are shown in Figure R14. It is clear that these particles remain unchanged after electrochemical cycling and there is no surface reconstruction to oxyhydroxides observed.

Figure R14. Identical location TEM (IL-TEM) images of the CoFe₂O₄ before (a, c, and e) and after OER measurement (b, d, and f).

References

- [1] V. Gatard et al., « Use of magnetic fields in electrochemistry: A selected review », *Curr. Opin. Electrochem.*, vol. 23, p. 96 105, oct. 2020, doi: 10.1016/j.coelec.2020.04.012.
- [2] L. M. A. Monzon et al., « Magnetically-Induced Flow during Electropolishing », *J. Electrochem. Soc.*, vol. 165, no 13, p. E679 E684, 2018, doi: 10.1149/2.0581813jes.
- [3] D. W. Banham et al. « Pt/Carbon Catalyst Layer Microstructural Effects on Measured and Predicted Tafel Slopes for the Oxygen Reduction Reaction », *J. Phys. Chem. C*, vol. 113, no 23, p. 10103 10111, juin 2009, doi: 10.1021/jp809987g.

REVIEWER COMMENTS

Reviewer #2 (Remarks to the Author):

I appreciate the concise summary of what distinguishes this manuscript from Ref [29] (Nature Energy 2019, 4, 519-525), but I still believe that the differences are only incremental. This manuscript shows a change in the Tafel slope, presents DFT results of questionable quality, the fact that the magnetic enhancement effect is not seen for other reactions, and the variation of the magnetic field strength. None of these highlighted points are truly novel.

Figure R2/S7: the bottom part showing the bond formation between O* and OH- shows the same spin alignment twice, but one is favorable, the other isn't. I believe one electron's spin should be flipped.

The response regarding the (111) surface termination was unsatisfactory. There are many more than just 2 possible terminations. For example, Zasada et al. has studied 9 different terminations and calculated a phase diagram as function of oxygen partial pressure and temperature [Zasada, F., Gryboś, J., Piskorz, W., & Sojka, Z. (2018). Cobalt Spinel (111) Facets of Various Stoichiometry - DFT+U and Ab Initio Thermodynamic Investigations. Journal of Physical Chemistry C, 122(5), 2866–2879. <https://doi.org/10.1021/acs.jpcc.7b11869>]. For the OER reaction, a Pourbaix diagram with dependence on the pH and applied voltage should be prepared. The surface termination under reaction conditions should be used as starting point, which will have implications on the "first" electron transfer step.

The computational hydrogen electrode and BEP relations are a great tool for catalyst screening, but they are not suitable to study a specific system or comparing elementary steps of different nature. Of the 4 OER steps, one is a simple binding event, two are proton transfer events, and one is an O-O bond formation step.

- (1) $\text{OH}^- + * \rightarrow * \text{OH} + \text{e}^-$;
- (2) $* \text{OH} + \text{OH}^- \rightarrow * \text{O} + \text{H}_2\text{O} + \text{e}^-$;
- (3) $* \text{O} + \text{OH}^- \rightarrow * \text{OOH} + \text{e}^-$;
- (4) $* \text{OOH} + \text{OH}^- \rightarrow * + \text{O}_2 + \text{H}_2\text{O} + \text{e}^-$.

It is a miscategorization to refer to all of these as OH- addition steps. The OH- addition in step (3) is the critical O-O bond formation step which warrants closer attention, as it was done, for example, in Xiao, Shin, Goddard III, PNAS 2018, <https://www.pnas.org/content/pnas/early/2018/05/17/1722034115.full.pdf> Without addressing the barrier change of this step in response to the magnetic field, this work isn't appropriate for Nature Communications.

The authors responded that "The octahedral and tetrahedral sites are antiferromagnetically coupled,...", but also stated that no AFM (antiferromagnetic) configurations were calculated. At least for Co₃O₄, the most stable AFM structure has the octahedral Co³⁺ ions with a zero magnetic moment, and the tetrahedral Co²⁺ ions in antiferromagnetic coupling.

Bajdich, M., García-Mota, M., Vojvodic, A., Nørskov, J. K., & Bell, A. T. (2013). Theoretical investigation of the activity of cobalt oxides for the electrochemical oxidation of water. Journal of the American Chemical Society, 135(36), 13521–13530. <https://doi.org/10.1021/ja405997s>

If the authors didn't find this AFM structure, then they likely should have provided suitable initial guesses instead of letting VASP figure it out. VASP is notoriously unreliable when it comes to identifying the preferred magnetic structure (for instance, it converges routinely to the O₂ singlet state, not the known triplet ground state).

If the spin structure of CoFe₂O₄ is not known, the authors should carefully investigate the sensitivity

to different initial moment guesses. Spinel oxides are infamous for strongly varying energies in different spin states.

Despite the many improvements that the authors have made on the manuscript, I remain skeptical that the results are anything more than just incremental. I also believe the DFT calculations are too simplistic to provide a rigorous and fundamental explanation for how quantum spin-exchange interactions accelerate the OER.

Reviewer #3 (Remarks to the Author):

The authors have added information with greater clarity to ensure the good reading of manuscript.

Comments

Figure 6 (g) shows the oscillation of current for demagnetization procedure. Please provide more information to explain this procedure: apparatus and corresponding magnetic field strength variations.

Please provide in supplementary material document, the figure R14. Identical location TEM (in rebuttal letter).

Reviewer #4 (Remarks to the Author):

This manuscript by Ren et al. reports the enhancement of the OER activity of the ferromagnetic oxide CoFe₂O₄ when this is exposed to an external magnetic field. This enhancement is attributed to the improved reaction kinetics due to spin selective electron transfer which favors the formation of triplet O₂. To support this, the authors show that the OER activity of the non-ferromagnetic oxides Co₃O₄ and IrO₂ do not benefit from this improvement. Tafel slope analysis indicates that the rate-determining step (RDS) for CoFe₂O₄ without the presence of a magnetic field corresponds to the *OH to O* transition, whereas the RDS under magnetization changes. To support and shed light on the experimental findings, DFT calculations are conducted on a CoFe₂O₄ (111) slab model showing that the RDS is the *OH to *O step. In all, the findings reported in this work might be of interest to the broad scientific community, however, there are several unsupported statements and major aspects (see below) which need to be addressed before I can recommend the publication of this manuscript.

General Comments:

1) In page 3, the authors state "the rate of a chemical reaction will be slow if the spin of the electronic wave function of the products differs from those of the reactants, as the Hamiltonian does not contain spin operators". Reference 17, however, states that the reaction rate is zero in that case, although this Reviewer has their reservations about both statements and does not think the reaction rate should necessarily be zero or lower. Unless the authors can provide further evidence to support this statement, I strongly encourage them to remove it.

2) The reduced enhancement of the magnetization as the temperature increases is attributed to the disturbance of the magnetic field. While this might be true, one should also note that CoFe₂O₄ has a more favorable Tafel slope as T increases. This could also be because chemical OER steps are favored over electrochemical steps at high-temperatures and low applied voltages, as discussed by Garcia-Melchor et al. (Nat. Commun. 2019. 10, 4993; ChemCatChem, 2016, 8, 1792).

3) Just because the Tafel slope for the non-FM catalysts does not change, it does not mean that the key step is the first ET. These non-FM materials could follow a different mechanism or follow the same

mechanism with different reaction kinetics. The authors should reconsider the statement at the end of page 10.

4) In my opinion, the model provided on page 12 adds nothing to the manuscript and the model and nomenclature employed are somewhat arbitrary. Firstly, the χ symbol employed to denote the spin part is typically used to denote a spin-orbital, composed of a spatial part and spin functions. Secondly, I assume that the $1/\sqrt{2}$ is the normalization factor of the wavefunction, but this should multiply all the terms. Authors should also note that the wavefunction should be antisymmetric to satisfy Pauli's exclusion principle. The terminology of antisymmetric parts used by the authors is rather misleading. Hence, I strongly recommend the authors to revise this model or remove it.

5) What is the energy difference between CoFe₂O₄ with and without spin alignment? Authors should also provide some support to the statement regarding the 3d-2p hybridization and the formation of ligand holes.

6) According to the authors, calculations were performed on the (111) surface of CoFe₂O₄ as this "is lowest in surface energy among other facets", but no further details are provided like the exact facets and their calculated surface energies.

7) There is also no mention in the manuscript which are the OER active sites, except in the responses to the Reviewers. This should be clarified in the main text. On this regard, authors should justify why octahedral Co sites were assumed as the active sites.

8) Before assessing the active sites, however, authors should justify (with data) the choice of the (111) surface termination. Details of the thickness of the modelled slabs, as well as the number of layers allowed to relax, should be provided. This Reviewer also noted that the recorded HRTEM images in Supplementary Figure 5 display the (400) and (220) before OER, and the facets (220) and (311) after OER. The authors should comment on this as well.

9) After addressing my comment above, the catalyst surface termination (resting state) should be assessed under relevant conditions. For this, the authors should compute the Pourbaix diagram of the CoFe₂O₄(111) surface, if this is the most representative surface in experiments. With the catalyst resting state, then all the possible OER active sites should be considered before extracting any conclusions.

10) Finally, most of the conclusions drawn in this work rely on the formation of a metal-oxyl intermediate and a "triplet" *OOH. In my opinion, the latter is not likely to display a triplet character, but the calculated spin densities for these OER intermediates should be able to confirm this or otherwise. Hence, these values should be reported and discussed in detail throughout the manuscript.

Minor comments:

1) The manuscript contains numerous grammatical errors and the English language usage in many sentences needs substantial work. I strongly recommend the authors to seek the help of a native English speaker, if one is available. Some examples of sentences that need to be addressed are:

- Page 3, line 64: the statement "the spin of the electronic potential on which the reaction occurs to result in more efficient oxygen production" should be reworded to make it understandable.
- Page 9, line 196: "The Tafel equation presents the relationship between where" should also be reworded.
- Page 9, line 197: "i₀" is not shown in the equation.
- Page 10, line 205: use either "about 120 mV*dec⁻¹" or "109 mV*dec⁻¹".
- Page 10, line 212: reword "This is because that the rate constant for reaction".
- Page 10, line 218: why are the Tafel slopes difference from those in Fig. 1f?
- Page 10, line 220: reword: "the positive influence of the magnetic field became not that remarkable under".

2) Page 12, line 263. It is not clear what the authors mean by spin alignment. The authors should

refer the reader to the Methods section for details.

3) Caption Fig. 5d: should not "AOR" be "EGOR"?

4) Page 18, line 357: the word "step" appears twice.

5) In the computational methods section the authors state that "The computational hydrogen electrode (CHE) model⁶⁴ was used to calculate the free energies of OER, based on which the free energy of an adsorbed species is defined as..." The CHE has nothing to do with the expression of ΔG , which is given by thermodynamics.

Response to the reviewers' comments

We would like to thank the reviewers for giving us comments, criticism, as well as valuable suggestions to our manuscript. We have revised the manuscript according to the reviewers' comments and all the changes are highlighted in red color in the revised manuscript. Below please find a point-to-point response to the reviewers' comments.

Reviewer #2 (Remarks to the Author):

I appreciate the concise summary of what distinguishes this manuscript from Ref [29] (Nature Energy 2019, 4, 519-525), but I still believe that the differences are only incremental. This manuscript shows a change in the Tafel slope, presents DFT results of questionable quality, the fact that the magnetic enhancement effect is not seen for other reactions, and the variation of the magnetic field strength. None of these highlighted points are truly novel.

Response: Thanks for the further comment. Respectively, we disagree with the reviewer's comments on the novelty. Our work clearly provided an important advance in experimental approaches for studying the magnetic field enhanced OER. These results are important, but none of them has been reported before.

Figure R2/S7: the bottom part showing the bond formation between O* and OH⁻ shows the same spin alignment twice, but one is favorable, the other isn't. I believe one electron's spin should be flipped.

Response: Thanks for pointing this out. We have corrected the mistake in the figure. The modifications (as shown in Figure R1) have been updated in Supplementary Figure 10 in the revised SI.

Figure R1 (Supplementary Figure 10). The production of the triplet intermediate O(↓)O(↓)H species.

The response regarding the (111) surface termination was unsatisfactory. There are many more than just 2 possible terminations. For example, Zasada et al. has studied 9 different terminations and calculated a phase diagram as function of oxygen partial pressure and temperature [Zasada, F., Gryboś, J., Piskorz, W., & Sojka, Z. (2018). Cobalt Spinel (111) Facets of Various Stoichiometry - DFT+U and Ab Initio Thermodynamic Investigations. Journal of

Physical Chemistry C, 122(5), 2866–2879. <https://doi.org/10.1021/acs.jpcc.7b11869>. For the OER reaction, a Pourbaix diagram with dependence on the pH and applied voltage should be prepared. The surface termination under reaction conditions should be used as starting point, which will have implications on the "first" electron transfer step.

Response: Thanks for the suggestions from the reviewer. We agree. We have calculated the Pourbaix diagram of CoFe_2O_4 as shown in Figure R2. We can see from the Pourbaix diagram that the surface termination of CoFe_2O_4 is oxygen termination under OER test conditions.

Figure R2 (Figure 4d in the revised manuscript). The calculated surface Pourbaix diagram of CoFe_2O_4 .

As suggested by the reviewer, the reaction started from the oxygen termination, i.e., between a ligand oxygen on the surface and the adsorbed oxygen species (OH^-), and the "first" electron transfer step is $\text{O}^* + \text{OH}^- \rightarrow \text{*OOH} + \text{e}^-$. The spin-sensitive electron transfer can reduce the barrier for the formation of the intermediate $\text{O}(\downarrow)\text{O}(\downarrow)\text{H}$ species (shown in Figure R3), for which the "first" electron transfer is promoted.

Figure R3 (Figure 4e in the revised manuscript). The spin polarization mechanism in OER starting from the step of $O^* + OH^- \rightarrow *OOH + e^-$.

To further improve the manuscript, the modifications have been added in Figure 4 in the revised manuscript. Related discussion is added in the revised manuscript (Line 255, Page 12 marked in red.)

“DFT calculations were performed to explore the different electronic structure of $CoFe_2O_4$ under an applied magnetic field (the computational details are shown in the Supplementary Information). As shown in the projected density of states (PDOS) of $CoFe_2O_4$ (Figure 4a), the 3d-2p hybridization of the $CoFe_2O_4$ become stronger after the spins are aligned. As compared with the $CoFe_2O_4$ with anti-parallel couplings, the $CoFe_2O_4$ with spin alignment has a higher spin density on the oxygen atoms (Figure 4b). The calculation indicates that the magnetic moment of the ligand hole in $CoFe_2O_4$ is $0.059 \mu_B$ without spin alignment and is $0.188 \mu_B$ with spin alignment, which indicates a FM ligand hole in $CoFe_2O_4$. A concomitant increment of the 3d-2p hybridization associate with FM ligand holes will facilitate spin-selected charge transport and optimize the kinetics of the spin-charge transfer in the three-phase interface.^{43, 58} Thus, the dominant FM exchange between the ferromagnetic catalyst and the adsorbed oxygen species (reactants) will happen (Figure 4c and Supplementary Figure 9) with smaller electron-electron repulsion, making that the first electron transfer is no longer the RDS. We further prepared the Pourbaix diagram of $CoFe_2O_4$ as shown in Figure 4d, which show that the surface termination of $CoFe_2O_4$ is oxygen termination under OER conditions. The reaction started between a ligand oxygen on the surface and the adsorbed oxygen species (OH^-), and the “first” electron transfer step is $O^* + OH^- \rightarrow *OOH + e^-$. The spin-related OER mechanism is shown in Figure 4e. The FM $CoFe_2O_4$ with FM ligand hole will form oxygen termination with fixed spin direction. The first electron transfer process led to the generation of $O(\downarrow)$. That is the first electron transfer step, which leads to a spin polarization process to form the triplet state intermediate $O(\downarrow)O(\downarrow)H$ species with a lower barrier (Supplementary Figure 10). Consequently, the triplet state intermediate $O(\downarrow)O(\downarrow)H$ species will prefer to generate the triplet state O_2 .”

Figure R4 (Figure 4 in the revised manuscript). Spin-polarized OER. (a) The projected density of states (PDOS) of CoFe₂O₄ without and with spin alignment. (b) The spin density of CoFe₂O₄ with and without spin alignment. (c) Schematic of spin-exchange mechanism for OER. The first electron transfer step is promoted by spin polarization through the FM exchange (QSEI), which gives smaller electronic repulsions and makes the adsorbed O species have a fixed spin direction. (d) The calculated Pourbaix diagram of the (111) surface of CoFe₂O₄. (e) The spin polarization mechanism of OER, starting from the step of O* + OH⁻ → *OOH + e⁻. (f) The free energy diagram of OER at 1.23 V (vs. RHE) with and without the spin alignment on the (111) surface of CoFe₂O₄ toward triplet oxygen generation.

The computational hydrogen electrode and BEP relations are a great tool for catalyst screening, but they are not suitable to study a specific system or comparing elementary steps of different nature. Of the 4 OER steps, one is a simple binding event, two are proton transfer events, and one is an O-O bond formation step.

- (1) $\text{OH}^- + * \rightarrow * \text{OH} + \text{e}^-$;
- (2) $* \text{OH} + \text{OH}^- \rightarrow * \text{O} + \text{H}_2\text{O} + \text{e}^-$;
- (3) $* \text{O} + \text{OH}^- \rightarrow * \text{OOH} + \text{e}^-$;
- (4) $* \text{OOH} + \text{OH}^- \rightarrow * + \text{O}_2 + \text{H}_2\text{O} + \text{e}^-$.

It is a miscategorization to refer to all of these as OH⁻ addition steps. The OH⁻ addition in step (3) is the critical O-O bond formation step which warrants closer attention, as it was done, for example, in Xiao, Shin, Goddard III, PNAS 2018, <https://www.pnas.org/content/pnas/early/2018/05/17/1722034115.full.pdf> Without addressing the barrier change of this step in response to the magnetic field, this work isn't appropriate for Nature Communications.

Response: Thanks for the comments and suggestions from the reviewer. We totally agree that the O-O bond formation step is very important for OER and the barriers calculation is a powerful tool to study the specify step of OER. In this work, we employed computational hydrogen electrode (CHE) model to explore the electrochemical proton-electron transfer during OER. In the calculated free energy diagram, the rate-determining step (RDS) is considered as the elementary step with the largest uphill free energy change. As shown in Figure 4f, the first step is RDS and the catalysts with aligned spins have a lower free energy to the reaction. Combined with Pourbaix diagram, we can confirm that the surface termination of CoFe₂O₄ is oxygen termination under OER conditions and the first step is indeed the O-O bond formation. As reported by Norskov et al. (J. Am. Chem. Soc. 2009, 131, 16, 5809–5815) that the kinetic barrier of two reaction states is proportional to the free energy gap of them, which is governed by the Bell–Evans–Polanyi (BEP) relation. Thus, it can be inferred that the O-O bond formation have a smaller barrier under an applied magnetic field.

In our study, the potential limiting step (PLS) is believed to be the deprotonation step of ligand OH on metal site which has the highest energetic barrier among four steps. However, we understand the reviewer's concern that the PLS is not necessarily the RDS, especially when considering the high kinetic barrier of O-O coupling. To address this, we consider these two steps (deprotonation step and the O-O coupling step) both potentially as the RDS to discuss the effect of spin alignment. In our manuscript, it was well discussed that the PLS (deprotonation step) is facilitated with aligned spins. Then, for the O-O coupling, it is also noted that the ΔG is lower with aligned spins. Under Bell–Evans–Polanyi (BEP) relation, this O-O coupling step should also have lower activation energy (kinetic barrier) with aligned spins. Similar BEP correlation can also be referred in recent paper such as (*Nature*. 2020, 587, 408-413). Overall, the DFT study has indicated that both energetic and kinetic barrier in OER can be reduced with an applied magnetic field.

The authors responded that "The octahedral and tetrahedral sites are antiferromagnetically coupled,..", but also stated that no AFM (antiferromagnetic) configurations were calculated. At least for Co₃O₄, the most stable AFM structure has the octahedral Co³⁺ ions with a zero magnetic moment, and the tetrahedral Co²⁺ ions in antiferromagnetic coupling.

Bajdich, M., García-Mota, M., Vojvodic, A., Nørskov, J. K., & Bell, A. T. (2013). Theoretical investigation of the activity of cobalt oxides for the electrochemical oxidation of water. *Journal of the American Chemical Society*, 135(36), 13521–13530. <https://doi.org/10.1021/ja405997s>

If the authors didn't find this AFM structure, then they likely should have provided suitable initial guesses instead of letting VASP figure it out. VASP is notoriously unreliable when it comes to identifying the preferred magnetic structure (for instance, it converges routinely to the O₂ singlet state, not the known triplet ground state).

If the spin structure of CoFe₂O₄ is not known, the authors should carefully investigate the sensitivity to different initial moment guesses. Spinel oxides are infamous for strongly varying energies in different spin states.

Response: Thanks for the comments. We agree that Co₃O₄ is AFM configuration as the reference reported (*J. Am. Chem. Soc.* 2013, 135, 13521–13530). The magnetic structure of Co₃O₄ have also been verified by neutron diffraction experiment, which revealed that Co₃O₄ has the octahedral Co³⁺ ions with no permanent moment, and the tetrahedral Co²⁺ ions have a moment of 3.26 μ_B with antiferromagnetic coupling at low temperature. But it should be noted that Co₃O₄ has a Neel temperature ($T_N \approx 40$ K) much lower than the room temperature (*J. Phys. Chem. Solids* 1964, 25, 1-10). When the temperature is greater than the Neel temperature, Co₃O₄ will be in paramagnetic state.

In this work, we used the extended X-ray absorption fine structure (EXAFS) to confirm the CoFe₂O₄ with an almost perfect inversed spinel structure, that is, the Fe³⁺ ions distributed almost equally in octahedral and tetrahedral sites and Co²⁺ ions mainly distributed in octahedral sites. We further calculated the effective magnetic moment (μ_{eff}) of CoFe₂O₄ to

be about 3.44 μ_B by Curie–Weiss fitting. The μ_{eff} for CoFe₂O₄ is very close to the idea inverse spinel value (*Journal of Magnetism and Magnetic Materials*, 1993, 123, 93-96). In inverse spinel CoFe₂O₄, the Fe³⁺ ions distribute equally in octahedral and tetrahedral sites with the spin in opposite direction, which do not contribute effective magnetic moment. Instead, the Co²⁺ ions in octahedral sites contribute to the ferromagnetic moments. Those results are also reported in previous experimental work (e.g., *Condensed Matter* 1988, 71, 193-197; *Phys. Rev. B* 1979, 19, 499).

Despite the many improvements that the authors have made on the manuscript, I remain skeptical that the results are anything more than just incremental. I also believe the DFT calculations are too simplistic to provide a rigorous and fundamental explanation for how quantum spin-exchange interactions accelerate the OER.

Reviewer #3 (Remarks to the Author):

The authors have added information with greater clarity to ensure the good reading of manuscript.

Comments

Figure 6 (g) shows the oscillation of current for demagnetization procedure. Please provide more information to explain this procedure: apparatus and corresponding magnetic field strength variations.

Response: Thanks for the comments. The electromagnet has been used in this work. The oscillation of current for demagnetization procedure is applied according to this formula:

$$I = I_0 e^{-0.012t} \sin\left(\frac{\pi}{3}t\right)$$

The magnetic field strength is proportional to the applied current as show in Figure R5.

Figure R5. The recorded demagnetization process of CoFe_2O_4 .

Please provide in supplementary material document, the figure R14. Identical location TEM (in rebuttal letter).

Response: Thanks for the suggestions. Figure R14 has been added in the revised SI as Supplementary Figure 11. Related methods are added in the revised manuscript (Line 64 Page S3 marked in red):

“We have carried out HRTEM observation on the CoFe_2O_4 catalyst at the identical location before and after the electrochemical reaction using identical location TEM technique (IL-TEM). The detailed methods of IL-TEM are as follows. A diluted ink containing catalyst was pipetted onto the gold finder grid (400 mesh, TED PELLA, USA). The pristine catalyst on the gold finder grid was observed at the specific location before cycling. Then the grid was used at the working electrode in an electrochemical cell. After the electrochemical cycling, the grid was dried under Ar flow and observed under TEM again. The grid allowed us to find the particle at the same location as it was before electrochemical cycling. The images are shown in Supplementary Figure 11. It is clear that those particles remain unchanged after

electrochemical cycling and there is no remarkable surface change.”

Figure R14 (Supplementary Figure 11 in the revised SI). Identical location TEM (IL-TEM) images of the CoFe₂O₄ before (a, c, and e) and after OER measurement (b, d, and f).

Reviewer #4 (Remarks to the Author):

This manuscript by Ren et al. reports the enhancement of the OER activity of the ferromagnetic oxide CoFe₂O₄ when this is exposed to an external magnetic field. This enhancement is attributed to the improved reaction kinetics due to spin selective electron transfer which favors the formation of triplet O₂. To support this, the authors show that the OER activity of the non-ferromagnetic oxides Co₃O₄ and IrO₂ do not benefit from this improvement. Tafel slope analysis indicates that the rate-determining step (RDS) for CoFe₂O₄ without the presence of a magnetic field corresponds to the *OH to O* transition, whereas the RDS under magnetization changes. To support and shed light on the experimental findings, DFT calculations are conducted on a CoFe₂O₄ (111) slab model showing that the RDS is the *OH to *O step. In all, the findings reported in this work might be of interest to the broad scientific community, however, there are several unsupported statements and major aspects (see below) which need to be addressed before I can recommend the publication of this manuscript.

General Comments:

1) In page 3, the authors state “the rate of a chemical reaction will be slow if the spin of the electronic wave function of the products differs from those of the reactants, as the Hamiltonian does not contain spin operators”. Reference 17, however, states that the reaction rate is zero in that case, although this Reviewer has their reservations about both statements and does not think the reaction rate should necessarily be zero or lower. Unless the authors can provide further evidence to support this statement, I strongly encourage them to remove it.

Response: Thanks for the comments and suggestions. We agree with the reviewer. We have removed this statement in the revised manuscript.

2) The reduced enhancement of the magnetization as the temperature increases is attributed to the disturbance of the magnetic field. While this might be true, one should also note that CoFe₂O₄ has a more favorable Tafel slope as T increases. This could also be because chemical OER steps are favored over electrochemical steps at high-temperatures and low applied voltages, as discussed by Garcia-Melchor et al. (Nat. Commun. 2019. 10, 4993; ChemCatChem, 2016, 8, 1792).

Response: Thanks for the reviewer’s kind reminder. It is true that we cannot ignore the mechanism change with increased temperature. It is possible that the I2M mechanism is also involved. The chemical step (i.e. the coupling of two M-O species) will be favorable under high temperature. We also learnt from the suggested refs (Nat. Commun. 2019. 10, 4993; ChemCatChem, 2016, 8, 1792) that the OER can have lower barrier under I2M than under AEM. That is also consistent with the changes of Tafel slope we observed at a high temperature. To be more rigorous, we have added these statements and cited those theoretical references in the revised manuscript. (Line 223, Page 10, marked in red):

“We also note that the Tafel slope of CoFe₂O₄ have a slight favorable change as temperature increases, which may be because the interaction between two M-O unites mechanism occurs at high temperature.^{53, 54”}

3) Just because the Tafel slope for the non-FM catalysts does not change, it does not mean that the key step is the first ET. These non-FM materials could follow a different mechanism or follow the same mechanism with different reaction kinetics. The authors should reconsider the statement at the end of page 10.

Response: Thanks for the reviewer's comment. We are sorry for the misleading statement. We have rephrased these sentences in the revised manuscript. (Line 225, Page 11, marked in red): "Thus, the key step in spin-polarized OER is the first electron transfer step in FM CoFe₂O₄, where the adsorbed OH⁻ is difficult to deprotonate and transfer the electron. However, the change of Tafel slopes was not observed in the non-ferromagnetic catalysts under the same condition."

4) In my opinion, the model provided on page 12 adds nothing to the manuscript and the model and nomenclature employed are somewhat arbitrary. Firstly, the chi symbol employed to denote the spin part is typically used to denote a spin-orbital, composed of a spatial part and spin functions. Secondly, I assume that the 1/sqrt(2) is the normalization factor of the wavefunction, but this should multiply all the terms. Authors should also note that the wavefunction should be antisymmetric to satisfy Pauli's exclusion principle. The terminology of antisymmetric parts used by the authors is rather misleading. Hence, I strongly recommend the authors to revise this model or remove it.

Response: Thanks for pointing this out. We fully agree with the reviewer's suggestion. We have removed this model and revised Figure 4.

Related discussion has been added in the revised manuscript (Line 255, Page 12 marked in red.)

"DFT calculations were performed to explore the different electronic structure of CoFe₂O₄ under an applied magnetic field (the computational details are shown in the Supplementary Information). As shown in the projected density of states (PDOS) of CoFe₂O₄ (Figure 4a), the 3d-2p hybridization of the CoFe₂O₄ become stronger after the spins are aligned. As compared with the CoFe₂O₄ with anti-parallel couplings, the CoFe₂O₄ with spin alignment has a higher spin density on the oxygen atoms (Figure 4b). The calculation indicates that the magnetic moment of the ligand hole in CoFe₂O₄ is 0.059 μB without spin alignment and is 0.188 μB with spin alignment, which indicates a FM ligand hole in CoFe₂O₄. A concomitant increment of the 3d-2p hybridization associate with FM ligand holes will facilitate spin-selected charge transport and optimize the kinetics of the spin-charge transfer in the three-phase interface.^{43, 58} Thus, the dominant FM exchange between the ferromagnetic catalyst and the adsorbed oxygen species (reactants) will happen (Figure 4c and Supplementary Figure 9) with smaller electron-electron repulsion, making that the first electron transfer is no longer the RDS. We further prepared the Pourbaix diagram of CoFe₂O₄ as shown in Figure 4d, which show that the surface termination of CoFe₂O₄ is oxygen termination under OER conditions. The reaction started between a ligand oxygen on the surface and the adsorbed oxygen species (OH⁻), and the "first" electron transfer step is O* + OH⁻ → *OOH + e⁻. The spin-related OER mechanism is shown in Figure 4e. The FM CoFe₂O₄ with FM ligand hole will form oxygen termination with

fixed spin direction. The first electron transfer process led to the generation of $O(\downarrow)$. That is the first electron transfer step, which leads to a spin polarization process to form the triplet state intermediate $O(\downarrow)O(\downarrow)H$ species with a lower barrier (Supplementary Figure 10). Consequently, the triplet state intermediate $O(\downarrow)O(\downarrow)H$ species will prefer to generate the triplet state O_2 ."

Figure R4 (Figure 4 in the revised manuscript). Spin-polarized OER. (a) The projected density of states (PDOS) of $CoFe_2O_4$ without and with spin alignment. (b) The spin density of $CoFe_2O_4$ with and without spin alignment. (c) Schematic of spin-exchange mechanism for OER. The first electron transfer step is promoted by spin polarization through the FM exchange (QSEI), which

gives smaller electronic repulsions and makes the adsorbed O species have a fixed spin direction. **(d)** The calculated Pourbaix diagram of the (111) surface of CoFe_2O_4 . **(e)** The spin polarization mechanism of OER starting from the step of $\text{O}^* + \text{OH}^- \rightarrow \text{*OOH} + \text{e}^-$. **(f)** The free energy diagram of OER at 1.23 V (vs. RHE) with and without the spin alignment on the (111) surface of CoFe_2O_4 toward triplet oxygen generation.

5) What is the energy difference between CoFe_2O_4 with and without spin alignment? Authors should also provide some support to the statement regarding the 3d-2p hybridization and the formation of ligand holes.

Response: Thanks for the suggestion. We have added the projected density of states (PDOS) of CoFe_2O_4 without and with spin alignment. As shown in Figure R6a, the 3d-2p hybridization of the CoFe_2O_4 become stronger after spin alignment. The calculation of spin density for CoFe_2O_4 show that the energy of CoFe_2O_4 slab model (oxygen-terminated) is ~ 505.22 eV, while the energy of the slab model with spin alignment is ~ 503.63 eV. The ligand hole in CoFe_2O_4 is bigger with spin alignment (as shown in Figure R6b). The magnetic moment of the ligand hole in CoFe_2O_4 is $0.059 \mu\text{B}$ without spin alignment and $0.188 \mu\text{B}$ with spin alignment. Those results show that the ligand hole in CoFe_2O_4 with spin alignment is ferromagnetic ligand hole.

Figure R6 (Figure 4a and b in the revised manuscript). **a**, The projected density of states (PDOS) of CoFe_2O_4 without and with spin alignment. **b**, The spin density for CoFe_2O_4 with and without spin alignment.

6) According to the authors, calculations were performed on the (111) surface of CoFe_2O_4 as this “is lowest in surface energy among other facets”, but no further details are provided like the exact facets and their calculated surface energies.

Response: Thanks for the comment. We are sorry that this statement is misleading. We chose (111) surface because the TEM investigation has shown that the surface of CoFe_2O_4 catalyst is rich in (111) surfaces. To further improve the manuscript, we have rephrased the statement in the revised manuscript. (Line 279, Page 13, marked in red):

“The (111) surface is chosen because the TEM investigation found the surface is rich in (111) and there is no remarkable change on the surface after OER (Supplementary Figure 11).”

7) There is also no mention in the manuscript which are the OER active sites, except in the responses to the Reviewers. This should be clarified in the main text. On this regard, authors should justify why octahedral Co sites were assumed as the active sites.

Response: Thanks for the suggestion. Previous experimental and computational work have shown that the preferential exposure of octahedral sites is a more general property of spinel oxides (Adv. Mater. 2017, 29, 1606800; Adv. Mater. 2019, 31, 1902509; *J. Phys. Chem. C* 2016, 120, 19087). Here, the extended X-ray absorption fine structure (EXAFS) showed the perfect inverse spinel structure of CoFe_2O_4 . Therefore, the Fe^{3+} ions distribute equally in octahedral and tetrahedral sites and Co^{2+} ions distribute in octahedral sites. We further calculated the effective magnetic moment (μ_{eff}) of CoFe_2O_4 to be about $3.44 \mu_B$ by Curie–Weiss fitting. The

μ_{eff} for CoFe_2O_4 is very close to the ideal value of the inverse spinel (Journal of Magnetism and Magnetic Materials, 1993, 123, 93-96). Thus, the Co^{2+} ions in octahedral sites contribute to the effective ferromagnetic moment. Those results are consistent in previous experimental work (Condensed Matter 1988, 71, 193-197). Considering that only Co in octahedral sites contribute the effective magnetic moment, the magnetic field enhanced OER should mainly happen on the Co sites. Thus, we studied the Co sites as the active sites in this work.

To further improve the manuscript, we have added statement about the active sites of CoFe_2O_4 in the revised manuscript and SI (Line 242, Page 12 in revised manuscript and Line 187, Page S10 in SI, marked in red):

“As revealed by our previous work, the octahedral sites are mainly responsible to the OER.⁵⁵ The extended X-ray absorption fine structure (EXAFS) showed the perfect inverse spinel structure of CoFe_2O_4 (Supplementary Figure 7). The Fe^{3+} ions distribute equally in octahedral and tetrahedral sites and Co^{2+} ions distribute in octahedral sites. We further calculated the effective magnetic moment (μ_{eff}) of CoFe_2O_4 to be about $3.44 \mu_B$ by Curie–Weiss fitting

(Supplementary Figure 8). The μ_{eff} for CoFe_2O_4 is very close to the ideal value of the inverse spinel.⁵⁶ Thus, the Co^{2+} ions in octahedral sites contribute to the effective ferromagnetic moment. Those results are consistent in previous experimental work.⁵⁷ Considering that only Co in octahedral sites contribute the effective magnetic moment, the magnetic field enhanced OER should mainly happen on the Co sites. Thus, we studied the Co sites as the active sites in this work.”

	Tetrahedral site	Octahedral site
Co_x	0.109	0.891
Co-O (Å)	1.81	2.06
Co-O coordination No.	3.3	4.9
Fe_y	0.891	1.109
Fe-O (Å)	1.93	1.99
Fe-O coordination No.	3.0	4.5
$\chi_w^2 = 376.9$; R factor = 0.0188		

Supplementary Figure 7. EXAFS $k^3\chi(R)$ spectra (gray circles) and fitting results (solid lines) of CoFe_2O_4 oxides at Co and Fe K-edge. The table shows the summary of EXAFS fitting results for CoFe_2O_4 . It confirms the cubic spinel structure of CoFe_2O_4 with ~90% Co cations in octahedral sites.

Supplementary Figure 8. The field-cooled M-T curves of CoFe₂O₄. The inset figure shows the temperature dependence of reciprocal susceptibilities. The solid line is the fitting results by the Curie–Weiss law. In the high temperature area, the susceptibilities derived from the magnetizations ($\chi = M/H$) obey a Curie–Weiss law: $\chi = C/T - T_C$, where C is Curie constant and T_C is Curie–Weiss temperature. By fitting the susceptibility versus and T data, an effective magnetic moment μ_{eff} can be obtained through $\mu_{eff} = \sqrt{8C} \mu_B$. Here, the calculated μ_{eff} of 3.44 μ_B for the CoFe₂O₄ sample is very close to the idea inverse spinel value.

8) Before assessing the active sites, however, authors should justify (with data) the choice of the (111) surface termination. Details of the thickness of the modelled slabs, as well as the number of layers allowed to relax, should be provided. This Reviewer also noted that the recorded HRTEM images in Supplementary Figure 5 display the (400) and (220) before OER, and the facets (220) and (311) after OER. The authors should comment on this as well.

Response: Thanks for the suggestions. The slab was modelled as a six-layer CoFe₂O₄ structure, containing 84 atoms. During the calculation, the top layer as well as the adsorbed OER intermediates were allowed to fully relax, while the bottom layers were kept fixed.

The prepared CoFe₂O₄ are polycrystalline powders, where all possible diffraction directions of the lattice should be attained due to the random orientation of the powdered material. However, we have found the surface is rich in (111) by TEM investigations. We have provided more representative TEM images to show the surface of particles in Supplementary Figure 5. We noted that the HRTEM images (Supplementary Figure 5) display the (400) and (220) before

OER, and the facets (220) and (311) after OER. We are sorry that we did not pick the most representative images in the previous version. In addition, the facet change is because of the images at different locations before and after OER. Here, we have carried out HRTEM to observe on the CoFe_2O_4 catalyst at the identical location before and after the electrochemical reaction using identical location TEM technique (IL-TEM). The images are shown in Figure R14.

Supplementary Figure 5. HRTEM images of CoFe_2O_4 before (a-c) and after OER (d, e). HRTEM images of Co_3O_4 before (f) and after OER (g). HRTEM images of IrO_2 before (h) and after OER (i).

Figure R14 (Supplementary Figure 11 in the revised SI). Identical location TEM (IL-TEM) images of the CoFe_2O_4 before (a, c, and e) and after OER measurement (b, d, and f).

9) After addressing my comment above, the catalyst surface termination (resting state) should be assessed under relevant conditions. For this, the authors should compute the Pourbaix diagram of the $\text{CoFe}_2\text{O}_4(111)$ surface, if this is the most representative surface in experiments. With the catalyst resting state, then all the possible OER active sites should be considered before extracting any conclusions.

Response: Thanks for the comments and suggestions. We have calculated the Pourbaix diagram of CoFe_2O_4 (as shown in Figure R7). We can see from Pourbaix diagram that the surface termination of CoFe_2O_4 is oxygen termination under OER test conditions (Figure R7).

Figure R7. The calculation model of CoFe_2O_4 for bulk (a) and surface (b). c, The calculated surface Pourbaix diagram of CoFe_2O_4 .

10) Finally, most of the conclusions drawn in this work rely on the formation of a metal-oxy intermediate and a “triplet” $^*\text{OOH}$. In my opinion, the latter is not likely to display a triplet character, but the calculated spin densities for the these OER intermediates should be able to confirm this or otherwise. Hence, these values should be reported and discussed in detail throughout the manuscript.

Response: Thanks for the suggestions from the reviewer. We have calculated spin densities of the adsorbed oxygen species and the calculation results are listed in the Table R1 below. The oxygen in metal-oxy intermediate ($^*\text{OOH}$) has the same direction spin, that is, $^*\text{OOH}$ is triplet.

Table R1 (Supplementary Table 4 in revised SI). Calculated spin densities (μ_B) of the adsorbed oxygen species.

	$^*\text{OH}$	$^*\text{O}$	$^*\text{OOH}$
Without spin alignment	0.065	0.195	O1: -0.025 O2: -0.063
With spin alignment	0.053	0.091	O1: -0.026 O2: -0.044

Minor comments:

1) The manuscript contains numerous grammatical errors and the English language usage in many sentences needs substantial work. I strongly recommend the authors to seek the help of a native English speaker, if one is available. Some examples of sentences that need to be addressed are:

– Page 3, line 64: the statement “the spin of the electronic potential on which the reaction occurs to result in more efficient oxygen production” should be reworded to make it understandable.

– Page 9, line 196: “The Tafel equation presents the relationship between where” should also be reworded.

– Page 9, line 197: “ i_0 ” is not shown in the equation.

– Page 10, line 205: use either “about 120 mV·dec⁻¹” or “109 mV·dec⁻¹”.

– Page 10, line 212: reword “This is because that the rate constant for reaction”.

– Page 10, line 218: why are the Tafel slopes difference from those in Fig. 1f?

– Page 10, line 220: reword: “the positive influence of the magnetic field became not that remarkable under”.

Response: We appreciate the reviewer for pointing out these issues. We have corrected them in the revised manuscript (marked in red).

Page 3, Lines 64:

“Ron Naaman and co-workers reported that the application of the chiral-induced spin selectivity effect to product the polarized electron. This spin polarization transferred is the origin of a more efficient oxidation process in which oxygen is formed in its triplet ground state.”

Page 9, line 196:

“The Tafel equation presents the relationship between the Tafel slope and the exchange current density:

$$\eta = -\frac{2.303 RT}{\alpha F} * \log i_0 + \frac{2.303 RT}{(\alpha + n)F} * \log i$$

where the Tafel slope equals to 2.303RT/[($\alpha+n$)F] (i_0 is the exchange current density, R is the universal gas constant, T is the absolute temperature, F is the Faraday constant, n is the number of electrons transferred before RDS, and α is the charge transfer coefficient and usually assumed to be 0.5).”

Page 9, line 197:

“ i_0 ” is shown in the Tafel equation. We are sorry that the Tafel equation is missing. We have added the equation in revised manuscripts.

Page 10, line 205:

For Tafel slope, the value of Tafel slope is 120 mV·dec⁻¹, which indicates the first electron transfer step is the RDS because there is no electron transfer before the RDS. If the second step is the RDS, the Tafel slope will decrease to 40 mV·dec⁻¹ with an electron transfer number of 1. In this work, the Tafel slope of CoFe₂O₄ is 109 mV·dec⁻¹ which is closed to 120 mV·dec⁻¹ and indicates the first electron transfer from the adsorbed OH⁻ is the RDS without the external

magnetic field.

Page 10, line 212:

“This is probably because that the rate constant of the reaction will increase as the reaction temperature increases, which can promote this reaction based on the transition state theory.”

Page 10, line 218:

Thanks for the questions. The two Tafel slope experiments are independent. But they are in the error bar range. Here, we added the error bar to those Tafel slope values in revised manuscript (Page 10, line 205 and 207, marked in red) to avoid confusion.

Page 10, line 220:

“As the temperature increases, the positive influence of the magnetic field became not remarkable.”

2) Page 12, line 263. It is not clear what the authors mean by spin alignment. The authors should refer the reader to the Methods section for details.

Response: Thanks for the suggestion from reviewer. We have added the reference in revised manuscript (Page 12, Line 248, marked in red).

3) Caption Fig. 5d: should not “AOR” be “EGOR”?

Response: Thanks for the comment. We have revised it to EGOR in the revised manuscript.

4) Page 18, line 357: the word “step” appears twice.

Response: Thanks for the comment. We have deleted it in the revised manuscript.

5) In the computational methods section the authors state that “The computational hydrogen electrode (CHE) model⁶⁴ was used to calculate the free energies of OER, based on which the free energy of an adsorbed species is defined as...” The CHE has nothing to do with the expression of ΔG , which is given by thermodynamics.

Response: Many thanks for pointing this out. This statement is indeed improper. We have changed the relative statements to “The computational hydrogen electrode (CHE) model was used to address the electrochemical proton-electron transfer with applied potential. The free energies of each elementary step were defined as...” (Line 90, Page S4, marked in red in the revised SI).

REVIEWER COMMENTS

Reviewer #4 (Remarks to the Author):

While the authors have successfully addressed many of my comments, some of the DFT data provided in the revisions do not support the main conclusions of this work. I, therefore, recommend that this paper is rejected until the authors come up with a reasonable mechanism for the enhancement of the OER activity with a magnetic field.

Comments

- 1) The caption in Figure 4 states that the energy diagram in 4f is calculated at 1.23 V vs RHE, but I believe this is incorrect as not all the steps should be uphill at 1.23 V. Intermediates in this figure should be labelled, as well.
- 2) In their reply to one of my comments, the authors state "... the energy of CoFe₂O₄ slab model (oxygen-terminated) is ~ 505.22 eV, while the energy of the slab model with spin alignment is ~503.63 eV." If this is correct, this means that the structure with spin alignment is 1.59 eV less stable, and therefore, it is very unlikely to form.
- 3) A stronger 3d-2p hybridization in CoFe₂O₄ after spin alignment is attributed based on the proximity of the lines (bands) in the PDOS plot shown in Figure 4a. Generally, one can safely assume that two atoms will interact if their bands overlap within the same range of energies, but the strength of this interaction cannot in principle be inferred from the proximity of the lines. Instead, one would need to do a more in-depth bonding analysis, for example, using crystal orbital Hamilton populations (COHP). Hence, the authors should remove any reference to the strength of that overlap inferred from the PDOS.
- 4) The authors state "... Co²⁺ ions in octahedral sites contribute to the effective ferromagnetic moment.". However, Fe³⁺ ions have also an odd number of electrons, and therefore, they should also contribute to the effective magnetic moment. Authors need to provide theoretical evidence that Co²⁺ ions are indeed the OER active sites and not Fe³⁺.
- 5) Relaxing only the topmost layer in the slab calculations is not sufficient to allow for surface relaxation upon the adsorption of the OER intermediates. At least 2-3 layers should be relaxed. The authors can assess the number of fixed layers needed by converging the surface energy with a different number of layers fixed. Authors should also note that catalysis does not always occur on the most exposed surface. The fact that the (220) and (311) facets appear after OER might indicate that the (111) is not the active phase. The authors should also provide a side view representation of the O-terminated surface slab used in the calculations.
- 6) Another of my biggest concerns with this manuscript are the calculated spin densities on the O atoms in the HOO* intermediate, which were not provided until now. These values are negligible (within 0.06 e⁻), and hence, they do not support the mechanistic hypothesis that the HOO* intermediate has one unpaired electron on each O atom. In addition, the sign of the spin densities on the O atoms is the same regardless of the spin alignment.

Response to reviewer's comments

Reviewer #4 (Remarks to the Author):

While the authors have successfully addressed many of my comments, some of the DFT data provided in the revisions do not support the main conclusions of this work. I, therefore, recommend that this paper is rejected until the authors come up with a reasonable mechanism for the enhancement of the OER activity with a magnetic field.

Respectively, we cannot agree with the reviewer's statement "some of the DFT data provided in the revisions do not support the main conclusions of this work" is confusing. We did not find out any evidence from the reviewer's comments below to support this statement.

Comments

1) The caption in Figure 4 states that the energy diagram in 4f is calculated at 1.23 V vs RHE, but I believe this is incorrect as not all the steps should be uphill at 1.23 V. Intermediates in this figure should be labelled, as well.

Response: This is a typical and established approach in literature. We can find many example papers doing this, including those papers published in Nat Comm (e.g., Nat. Commun 2020, 11, 2522, Nat. Energy 2016, 1, 16053, Nat. Commun 2020, 11, 1378, etc.). We have cited two of these references for reader's information (Ref. 61 and 62 in the revised manuscript).

2) In their reply to one of my comments, the authors state "... the energy of CoFe₂O₄ slab model (oxygen-terminated) is ~ 505.22 eV, while the energy of the slab model with spin alignment is ~503.63 eV." If this is correct, this means that the structure with spin alignment is 1.59 eV less stable, and therefore, it is very unlikely to form.

Response: We address below the concern on this comment. First of all, this OER reaction is carried out under a magnetic field. The equilibrium state of CoFe₂O₄ under a magnetic field must be different from the equilibrium state without magnetic field. Second, the CoFe₂O₄ slab model we calculated contains 84 atoms. The electronic energy of the structure with spin alignment is 1.59 eV higher than that of the structure without spin alignment. The energy change of each atom in the structure with/without spin alignment is only 0.0189 eV, which is a reasonable value. Lastly, if we look at the energy with and without spin alignment, there is only <0.4% difference. It cannot tell the spin aligned situation "very unlikely to form".

3) A stronger 3d-2p hybridization in CoFe₂O₄ after spin alignment is attributed based on the proximity of the lines (bands) in the PDOS plot shown in Figure 4a. Generally, one can safely assume that two atoms will interact if their bands overlap within the same range of energies, but the strength of this interaction cannot in principle be inferred from the proximity of the lines. Instead, one would need to do a more in-depth bonding analysis, for example, using crystal orbital Hamilton populations (COHP). Hence, the authors should remove any reference to the strength of that

overlap inferred from the PDOS.

Response: This is a typical approach in literature (e.g., Nat Commun 2018, 9, 4597, Nature Mater 2019, 18, 256–265, etc.). The current DFT approach is good enough as it is a “benchmarked” approach in literatures. We have added explanation in the manuscript and cited the reference to support (Ref.58 in the revised manuscript).

4) The authors state “... Co²⁺ ions in octahedral sites contribute to the effective ferromagnetic moment.”. However, Fe³⁺ ions have also an odd number of electrons, and therefore, they should also contribute to the effective magnetic moment. Authors need to provide theoretical evidence that Co²⁺ ions are indeed the OER active sites and not Fe³⁺.

Response: We have explained “effective” moment in our last round response. For OER, Co site is indeed much more active than Fe in spinel oxides, which has been well-reported in literatures (e.g., Adv. Mater. 2018, 30, 1802912, Nature Catalysis 2019, 2, 763-772, etc.). In addition, the Fe³⁺ in octahedral sites are present as Fe³⁺ ($t_{2g}^3e_g^2$) in the spinel CoFe₂O₄, which binds the oxygen too weakly and fail to activate the reactant based on the e_g orbital occupancy principle (Science 2011, 334, 1383-1385). Having an odd number of electrons does not mean being responsible to the expressed magnetic property of an oxide. The basic knowledge about the magnetism of spinel ferrites can be found in textbook of magnetism (Gerald F. Dionne, Magnetic Oxides, Springer Science+Business Media, USA, 2009, ISBN 978-1-4419-0053-1, Section 4). And we actually have explained it in the last round revision (it does not contribute to effective moment).

5) Relaxing only the topmost layer in the slab calculations is not sufficient to allow for surface relaxation upon the adsorption of the OER intermediates. At least 2-3 layers should be relaxed. The authors can assess the number of fixed layers needed by converging the surface energy with a different number of layers fixed. Authors should also note that catalysis does not always occur on the most exposed surface. The fact that the (220) and (311) facets appear after OER might indicate that the (111) is not the active phase. The authors should also provide a side view representation of the O-terminated surface slab used in the calculations.

Response: Respectively, we disagree with the reviewer. No evidence found in literature (e.g., Nat Commun 2018, 9, 3202, Nature Mater 2006, 5, 909–913) that at least 2- 3 layers “should” be relaxed. In addition, we did perform the surface energy calculations of the slab model with one layer and two layers to be fully relaxed, respectively. Based on the results, the surface energy of one-layer-relaxed model is 0.417 eV Å⁻¹; while that of the two-layer-relaxed model is 0.411 eV Å⁻¹. Considering the fact that the surface energies of these two models are pretty close, we therefore used one-layer-relaxed model for the following calculations to save the computational resources. We have explained to readers and cited the approach in the revised manuscript to support (Ref.60).

“The fact that the (220) and (311) facets appear after OER might indicate that the (111) is not the active phase.” There is NO such a fact in our manuscript. The IL-TEM (Supplementary Figure 11) has shown no surface change after OER.

The side view representation of the O-terminated surface slab used in the calculations is shown below (Figure R1). It has been added into the SI as Supplementary Figure 14.

Figure R1. The calculation model of CoFe_2O_4 with side view for bulk (a) and surface (b). It has been added into the SI. (Supplementary Figure 14.)

6) Another of my biggest concerns with this manuscript are the calculated spin densities on the O atoms in the HOO^* intermediate, which were not provided until now. These values are negligible (within $0.06 e^-$), and hence, they do not support the mechanistic hypothesis that the HOO^* intermediate has one unpaired electron on each O atom. In addition, the sign of the spin densities on the O atoms is the same regardless of the spin alignment.

Response: To provide the calculated spin densities on the O atoms was just asked in the last round when this reviewer came in. And we provided it in the revision. It should not be a problem now.

$0.06 e^-$ is not negligible. It should not be a big number. Please be noted that both situations produce triplet oxygen. Compared to the ground state triplet oxygen, excited singlet oxygen is about 1 eV higher in energy. If we follow the logic of the reviewer, the control one should give singlet oxygen, which will be very wrong.

Please note that there is NO hypothesis that the HOO^* intermediate having one unpaired electron on each O atom in our manuscript.

The reviewer assumed that “the value of spin density (0.06) are negligible”. However, please note that such assumption was only made by speculation and is unfounded. There is no evidence to support a criterion for such judgement. If the HOO^* intermediate is completely in singlet state with all electrons paired, why this value is not “0”? It is obviously that the spin character in ligand oxygens cannot be completely ignored, which is significant for the spin-related kinetics in OER. Following the logic of reviewer that the HOO^* intermediate has only paired electrons, the subsequent ground-state O_2 ($\uparrow\text{O}=\text{O}\uparrow$) turnover will need spin flip, which will encounter high energetic/kinetic barrier. Such high-barrier step should definitely not be involved in a favorable pathway toward O_2 production. Moreover, we also did not make a hypothesis that every oxygen in

HOO* intermediate has "one electron". The reviewer has mixed up our arguments. At last, the reviewer argued that the sign of the spin densities on the O atoms do not indicate the spin direction. It is fundamentally incorrect. The sign of spin density can only indicate the spin direction. In AFM coupled structure, spins are in antiparallel alignment and the sign of spin density will be the opposite. In FM coupled structure, the spins are in parallel spin alignment and the sign will be the same.